# PRE-TRAINING AND FINE-TUNING GENERATIVE FLOW NETWORKS

**Ling Pan**[1,2]**, Moksh Jain**[2,3]**, Kanika Madan**[2,3]**, Yoshua Bengio**[2,3,4]
[1]Hong Kong University of Science and Technology
[2]Mila - Québec AI Institute [3]Université de Montréal [4]CIFAR AI Chair
{penny.ling.pan}@gmail.com

## ABSTRACT

Generative Flow Networks (GFlowNets) are amortized samplers that learn stochastic policies to sequentially generate compositional objects from a given unnormalized reward distribution. They can generate diverse sets of high-reward objects, which is an important consideration in scientific discovery tasks. However, as they are typically trained from a given extrinsic reward function, it remains an important open challenge about how to leverage the power of pre-training and train GFlowNets in an unsupervised fashion for efficient adaptation to downstream tasks. Inspired by recent successes of unsupervised pre-training in various domains, we introduce a novel approach for reward-free pre-training of GFlowNets. By framing the training as a self-supervised problem, we propose an outcome-conditioned GFlowNet (OC-GFN) that learns to explore the candidate space. Specifically, OC-GFN learns to reach any targeted outcomes, akin to goal-conditioned policies in reinforcement learning. We show that the pre-trained OC-GFN model can allow for a direct extraction of a policy capable of sampling from any new reward functions in downstream tasks. Nonetheless, adapting OC-GFN on a downstream task-specific reward involves an intractable marginalization over possible outcomes. We propose a novel way to approximate this marginalization by learning an amortized predictor enabling efficient fine-tuning. Extensive experimental results validate the efficacy of our approach, demonstrating the effectiveness of pre-training the OC-GFN, and its ability to swiftly adapt to downstream tasks and discover modes more efficiently. This work may serve as a foundation for further exploration of pre-training strategies in the context of GFlowNets.

## 1 INTRODUCTION

Unsupervised learning on large stores of data on the internet has resulted in significant advances in a variety of domains (Howard & Ruder, 2018; Devlin et al., 2018; Radford et al., 2019; Henaff, 2020). Pre-training with unsupervised objectives, such as next-token prediction in auto-regressive language models (Radford et al., 2019), on large-scale unlabelled data enables the development of models that can be effectively fine-tuned for novel tasks using few samples (Brown et al., 2020). Unsupervised learning at scale allows models to learn good representations, which enables data-efficient adaptation to novel tasks, and is central to the recent development towards larger models.

On the other hand, in the context of amortized inference, Generative Flow Networks (GFlowNets; Bengio et al., 2021) enable learning generative models for sampling from high-dimensional distributions over discrete compositional objects. Inspired by reinforcement learning (RL), GFlowNets learn a stochastic policy to sequentially generate compositional objects with a probability proportional to a given reward, instead of reward maximization. Therefore, GFlowNets have found success in applications to scientific discovery problems to generate high-quality and diverse candidates (Bengio et al., 2021; Jain et al., 2023a;b) as well as alternatives to Monte-Carlo Markov chains and variational inference for modeling Bayesian posteriors (Deleu et al., 2022; van Krieken et al., 2022).

As a motivating example, consider the drug discovery pipeline. A GFlowNet can be trained to generate candidate RNA sequences that bind to a target using as reward the binding affinity of the RNA sequence with the target (Lorenz et al., 2011; Sinai et al., 2020) (which can be uncertain and imperfect based on the current understanding of the biological system and available experimental data). However, there is no way to efficiently adapt the GFlowNet to sample RNA sequences binding

to a different target of interest that reflects new properties. Unlike human intelligence, GFlowNets currently lack the ability to leverage previously learned knowledge to efficiently adapt to new tasks with unseen reward functions, and need to be trained from scratch to learn a policy for matching the given extrinsic reward functions for different tasks. Inspired by the success of the unsupervised pre-training and fine-tuning paradigm in vision, language, and RL (Jaderberg et al., 2016; Sekar et al., 2020) domains, it is natural to ask how can this paradigm benefit GFlowNets. As a step in this direction, we propose a fundamental approach to realize this paradigm for GFlowNets.

In this paper, we propose a novel method for reward-free unsupervised pre-training of GFlowNets. We formulate the problem of pre-training GFlowNets as a self-supervised problem of learning an outcome-conditioned GFlowNet (OC-GFN) which learns to reach any outcome (goal) as a functional understanding of the environment (akin to goal-conditioned RL (Chebotar et al., 2021)). The reward for training this OC-GFN is defined as the success of reaching the outcome. Due to the inherent sparse nature of this task-agnostic reward, it introduces critical challenges for efficient training OC-GFN in complex domains, particularly with higher-dimensional outcomes and long-horizon problems, since it is difficult to reach the outcome to get a meaningful reward signal. To tackle these challenges, we introduce a novel contrastive learning procedure to train OC-GFN to effectively handle such sparse rewards, which induces an implicit curriculum for efficient learning that resembles goal relabeling (Andrychowicz et al., 2017). To enable efficient learning in long-horizon tasks, we further propose a goal teleportation scheme to effectively propagate the learning signal to each step.

A remarkable result is that one can directly convert this pre-trained OC-GFN to sample proportionally to a new reward function for downstream tasks (Bengio et al., 2023). It is worth noting that in principle, this can be achieved even without re-training the policy, which is usually required for fine-tuning in RL, as it only learns a reward-maximizing policy that may discard many useful information. Adapting the pre-trained OC-GFN model to a new reward function, however, involves an intractable marginalization over possible outcomes. We propose a novel alternative by learning a predictor that amortizes this marginalization, allowing efficient fine-tuning of the OC-GFN to downstream tasks. Our key contributions can be summarized as follows:

- We propose reward-free pre-training for GFlowNets as training outcome-conditioned GFlowNet (OC-GFN) that learns to sample trajectories to reach any outcome.

- We investigate how to leverage the pre-trained OC-GFN model to adapt to downstream tasks with new rewards, and we also introduce an efficient method to learn an amortized predictor to approximate an intractable marginal required for fine-tuning the pre-trained OC-GFN model.

- Through extensive experiments on the GridWorld domain, we empirically validate the efficacy of the proposed pre-training and fine-tuning paradigm. We also demonstrate its scalability to larger-scale and challenging biological sequence design tasks, which achieves consistent and substantial improvements over strong baselines, especially in terms of diversity of the generated samples.

## 2 PRELIMINARIES

Given a space of compositional objects $\mathcal{X}$, and a non-negative reward function $R : \mathcal{X} \mapsto \mathbb{R}^+$, the GFlowNet policy $\pi$ is trained towards sampling objects $x \in \mathcal{X}$ from the distribution defined by $R(x)$, i.e., $\pi(x) \propto R(x)$. The compositional objects are each sampled sequentially, with each step involving the addition of a building block $a \in \mathcal{A}$ (action space) to the current partially constructed object $s \in \mathcal{S}$ (state space). We can define a directed acyclic graph (DAG) $\mathcal{G} = \{\mathcal{S}, \mathcal{A}\}$ with the partially constructed objects forming nodes of $\mathcal{G}$, including a special empty state $s_0$. The edges are $s \rightarrow s'$, where $s'$ is obtained by applying an action $a \in \mathcal{A}$ to $s$. The complete objects $\mathcal{X}$ are the terminal (childless) nodes in the DAG. The generation of an object $x \in \mathcal{X}$ corresponds to complete trajectories in the DAG starting from $s_0$ and terminating in a terminal state $s_n = x$, i.e., $\tau = (s_0 \rightarrow \cdots \rightarrow x)$. We assign a non-negative weight, called *state flow* $F(s)$ to each state $s \in \mathcal{S}$. The forward policy $P_F(s'|s)$ is a collection of distributions over the children of each state and the backward policy $P_B(s|s')$ is a collection of distributions over the parents of each state. The forward policy induces a distribution over trajectories $P_F(\tau)$, and the marginal likelihood of sampling a terminal state is given by marginalizing over trajectories terminating in $x$, $P_F^\top(x) = \sum_{\tau=(s_0 \rightarrow \cdots \rightarrow x)} P_F(\tau)$. GFlowNets solve the problems of learning a parameterized policy $P_F(\cdot \mid \cdot; \theta)$ such that $P_F^\top(x) \propto R(x)$.

**Training GFlowNets.** We use the *detailed balance* learning objective (DB; Bengio et al., 2023) to learn the parameterized policies and flows based on Eq. (1), which considers the flow consistency constraint in the edge level (i.e., the incoming flow for edge $s \rightarrow s'$ matches the outgoing flow).

When it is trained to completion, the objective yields the desired policy.

$$\forall s \rightarrow s' \in \mathcal{A}, \qquad F(s)P_F(s'|s) = F(s')P_B(s|s'). \tag{1}$$

As in reinforcement learning, exploration is a key challenge in GFlowNets. Generative Augmented Flow Networks (GAFlowNets; Pan et al., 2023b) incorporate intrinsic intermediate rewards represented as augmented flows in the flow network to drive exploration, where $r_i(s \rightarrow s')$ is specified by intrinsic motivation (Burda et al., 2018), yielding the following variant of detailed balance.

$$\forall s \rightarrow s' \in \mathcal{A}, \qquad F(s)P_F(s'|s) = F(s')P_B(s|s') + r(s \rightarrow s'). \tag{2}$$

## 3 RELATED WORK

**GFlowNets.** While several learning objectives have been proposed for improving credit assignment and sample efficiency in GFlowNets (Bengio et al., 2021; 2023) such as detailed balance (Bengio et al., 2023), sub-trajectory balance (Madan et al., 2022) and forward-looking objectives (Pan et al., 2023a), they need to be trained from scratch with a given reward function, which may limit its applicability to more practical problems. Owing to their flexibility, GFlowNets have been applied to wide range of problems where diverse high-quality candidates are need, such as molecule generation (Bengio et al., 2021), biological sequence design (Jain et al., 2022), combinatorial (Zhang et al., 2023a;b), and multi-objective optimization (Jain et al., 2023b). There have also been recent efforts in generalizing GFlowNets to handle stochastic environments including Stochastic GFlowNets (Pan et al., 2023c) and Distributional GFlowNets (Zhang et al., 2023c), which are effective in handling stochasticity in transition dynamics and rewards; and also improving training efficiency of GFlowNets inspired by reinforcement learning (Pan et al., 2020; Lau et al., 2024) and evolutionary algorithms (Ikram et al., 2024).

**Unsupervised Pre-Training in Reinforcement Learning.** Following the progress in language modeling and computer vision, there has been growing interest in pre-training reinforcement learning (RL) agents in an unsupervised stage without access to task-specific rewards for learning representations. Agents typically learn a set of different skills (Eysenbach et al., 2018; Hansen et al., 2019; Zhao et al., 2021; Liu & Abbeel, 2021), and then fine-tune the learned policy to downstream tasks. Contrary to reward-maximization, GFlowNets learn a stochastic policy to match the reward distribution. As we show in the next section, such a learned policy can be adapted to a new reward function even without re-training (Bengio et al., 2023).

**Goal-Conditioned Reinforcement Learning.** Different from standard reinforcement learning (RL) methods that learn policies or value functions based on observations, goal-conditioned RL (Kaelbling, 1993) also take goals into consideration by augmenting the observations with an additional input of the goal and have been studied in a number of prior works (Schaul et al., 2015; Nair et al., 2018; Veeriah et al., 2018; Eysenbach et al., 2020). Goal-conditioned RL is trained to greedily achieve different goals specified explicitly as input, making it possible for agents to generalize their learned abilities across different environments.

## 4 PRE-TRAINING AND FINE-TUNING GENERATIVE FLOW NETWORKS

In the original formulation, a GFlowNet need to be trained from scratch whenever it is faced with a previously unseen reward function (with consistent state and action spaces, as motivated in Section 1). In this section, we aim to leverage the power of pre-training in GFlowNets for efficient adaptation to downstream tasks. To tackle this important challenge, we propose a novel approach to frame the problem of pre-training GFlowNets as a self-supervised problem, by training an outcome-conditioned GFlowNet that learns to reach any input terminal state (outcome) without task-specific rewards. Then, we propose to leverage the power of

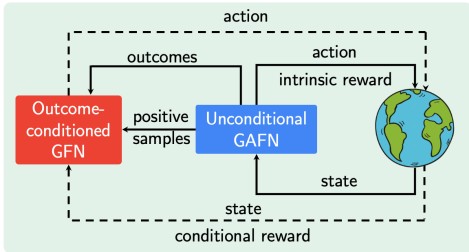

Figure 1: The unsupervised pre-training phase of outcome-conditioned GFlowNet.

the pre-trained GFlowNet model for efficiently fine-tuning it to downstream tasks with a new reward function.

### 4.1 UNSUPERVISED PRE-TRAINING STAGE

As GFlowNets are typically learned with a given reward function, it remains an open challenge for how to pre-train them in a reward-free fashion. We propose a novel method for unsupervised

---

**Algorithm 1** Reward-free Pre-training of Unsupervised GFlowNets.

---

**Require:** GAFlowNet $F^U(s)$, $P_F^U(s'|s)$, $P_B^U(s|s')$; fixed target network $\bar{\phi}$; predictor network $\phi$;
    Outcome-conditioned GFlowNet (OC-GFN) $F^C(s|y)$, $P_F^C(s'|s, y)$, $P_B^C(s|s', y)$
1: **for** each training step $t = 1$ to $T$ **do**
2:     Collect a trajectory $\tau^+ = \{s_0^+ \rightarrow \cdots \rightarrow s_n^+\}$ with $P_F^U$
3:     *// Update the outcome-conditioned GFlowNet model*
4:     $y^+ \leftarrow f(s^+)$, $R(s_n^+|y^+) \leftarrow \mathbb{1}\{f(s_n^+) = y^+\} \equiv 1$
5:     Update OC-GFN towards minimizing Eq. (5) with $\tau^+$ and $R(s_n^+|y^+)$
6:     Collect a trajectory $\tau^- = \{s_0^- \rightarrow \cdots \rightarrow s_n^-\}$ with $P_F^C$ conditioned on $y^+$
7:     $y^- \leftarrow f(s^-)$, $R(s_n^-|y^+) \leftarrow \mathbb{1}\{f(s_n^-) = y^+\}$
8:     Update OC-GFN towards minimizing Eq. (5) with $\tau^-$ and $R(s_n^-|y^+)$
9:     *// Update the unconditional GAFlowNet model*
10:    Update GAFlowNet towards optimizing Eq. (2) based on $\tau^+$ and $r_i \leftarrow ||\bar{\phi}(s) - \phi(s)||_2$
11:    Update the predictor network $\phi$ towards minimizing $||\bar{\phi}(s) - \phi(s)||_2$

---

pre-training of GFlowNets without task-specific rewards. We formulate the problem of pre-training GFlowNets as a self-supervised problem of learning an outcome-conditioned GFlowNet (OC-GFN) which learns to reach any input target outcomes, inspired by the success of goal-conditioned reinforcement learning in generalizing to a variety of tasks when it is tasked with different goals (Fang et al., 2022). We denote the outcome as $y = f(s)$, where $f$ is an identity function and $s$ is a terminal state, so the space of outcomes is the same as the state space $\mathcal{X}$. What makes OC-GFN special is that, when fully trained, given a reward $R$ a posterior as a function $r$ of the outcome, i.e., $R(s) = r(y)$, one can adapt the OC-GFN to sample from this reward, which can generalize to tasks with different rewards.

### 4.1.1 OUTCOME-CONDITIONED GFLOWNETS (OC-GFN)

We extend the idea of flow functions and policies in GFlowNets to OC-GFN that can generalize to different outcomes $y$ in the outcome space, which is trained to achieve specified $y$. OC-GFN can be realized by providing an additional input $y$ to the flows and policies, resulting in outcome-conditioned forward and backward policies $P_F(s'|s, y)$ and $P_B(s|s', y)$, and flows $F(s|y)$. The resulting learning objective for OC-GFN for intermediate states is shown in Eq. (3).

$$F(s|y)P_F(s'|s, y) = F(s'|y)P_B(s|s', y). \tag{3}$$

**Outcome generation** We can train OC-GFN by conditioning outcomes-conditioned flows and policies on a specified outcome $y$, and we study how to generate them autotelically. It is worth noting that we need to train it with full-support over $y$. We propose to leverage GAFN (Pan et al., 2023b) with augmented flows that enable efficient reward-free exploration purely by intrinsic motivation (Burda et al., 2018). In practice, we generate diverse outcomes $y$ with GAFN, and provide them to OC-GFN to sample an outcome-conditioned trajectory $\tau = (s_0, \cdots, s_n)$. The effect of the GAFN is studied in Appendix C.2, which is critical in large and high-dimensional problems as it affects the efficiency of generating diverse outcomes. The resulting conditional reward is given as $R(s_n|y) = \mathbb{1}\{f(s_n) = y\}$. Thus, OC-GFN receives a zero reward if it fails to reach the target outcome, and a positive reward otherwise, which results in learning an outcome-achieving policy.

**Contrastive training** However, it can be challenging to efficiently train OC-GFN in problems with large outcome spaces. This is because it can be hard to actually achieve the outcome and obtain a meaningful learning signal owing to the sparse nature of the rewards — most of the conditional rewards $R(s_n|y)$ will be zero if $f(s_n) \neq y$ when OC-GFN fails to reach the target.

To alleviate this we propose a contrastive learning objective for training the OC-GFN. After sampling a trajectory $\tau^+ = \{s_0^+ \rightarrow \cdots \rightarrow s_n^+\}$ from unconditional GAFN, we first train an OC-GFN based on this off-policy trajectory by assuming it has the ability to achieve $y^+ = s_n^+$ when conditioned on $y^+$. Note that the resulting conditional reward $R(s_n^+|y^+) = \mathbb{1}\{f(s_n^+) = y^+\} \equiv 1$ in this case, as all correspond to successful experiences that provide meaningful learning signals to OC-GFN. We then sample another on-policy trajectory $\tau^- = \{s_0^- \rightarrow \cdots \rightarrow s_n^-\}$ from OC-GFN by conditioning it on $y^+$, and evaluate the conditional reward by $R(s_n^-|y^+) = \mathbb{1}\{f(s_n^-) = y^+\}$. Although most of $R(s_n^-|y^+)$ can be zero in large outcome spaces during early learning, we provide sufficient successful experiences in the initial phase, which is also related to goal relabeling (Andrychowicz et al., 2017). This can be viewed as an implicit curriculum for improving the training of OC-GFN.

---

**Algorithm 2** Supervised Fine-Tuning of Outcome-Conditioned GFlowNets.

---

1: Initialize the numerator network $N(s'|s)$ and the GFlowNet-like predictor network $Q(y|s', s)$
2: Obtain the pre-trained outcome-conditioned state $F(s|y)$ and forward policy $P_F(s'|s, y)$
3: **for** each training step $t = 1$ to $T$ **do**
4:     Collect a trajectory $\tau = \{s_0 \to \cdots s_n\}$ with $N(s'|s)$
5:     Sample outcomes $y$ from a tempered/$\epsilon$-greedy version of $Q(\cdot|s', s)$
6:     Update $N$ and $Q$ towards minimizing Eq. (9)
7: Compute the policy $P_F^r(s'|s) = N(s'|s)/\sum_{s'} N(s'|s)$

---

**Outcome teleportation**   The contrastive training paradigm can significantly improve learning efficiency by providing a bunch of successful trajectories with meaningful learning signals, which tackles the particular challenge of sparse rewards during learning. However, the agent may still suffer from poor learning efficiency in long-horizon tasks, as it cannot effectively propagate the success/failure signal back to each step. We propose a novel technique, *outcome teleportation*, for further improving the learning efficiency of OC-GFN as in Eq. (4), which considers the terminal reward $R(x|y)$ for every transition (noting the binary nature of the rewards $R$).

$$F(s|y)P_F(s'|s, y) = F(s'|y)P_B(s|s', y)R(x|y). \tag{4}$$

This formulation can efficiently propagate the guidance signal to each transition in the trajectory, which can significantly improve learning efficiency, particularly in high-dimensional outcome spaces, as investigated in Section 5.2. It can be interpreted as a form of reward decomposition in outcome-conditioned tasks with *binary* rewards. In practice, we train OC-GFN by minimizing the following loss function $\mathcal{L}_{OC-GFN}(\tau, y)$ in log-scale obtained from Eq. (4), i.e.,

$$\sum_{s \to s' \in \tau} \left(\log F(s|y) + \log P_F(s'|s, y) - \log F(s'|y) - \log P_B(s|s', y) - \log R(x|y)\right)^2. \tag{5}$$

**Theoretical justification.** We now justify that when OC-GFN is trained to completion, it can successfully reach any specified outcome. The proof can be found in Appendix B.1.

**Proposition 4.1.** *If $\mathcal{L}_{OC-GFN}(\tau, y) = 0$ for all trajectories $\tau$ and outcomes $y$, then the outcome-conditioned forward policy $P_F(s'|s, y)$ can successfully reach the target outcome $y$.*

The resulting procedure for the reward-free unsupervised pre-training of OC-GFN is summarized in Algorithm 1 and illustrated Figure 1.

### 4.2   SUPERVISED FINE-TUNING STAGE

In this section, we study how to leverage the pre-trained OC-GFN model and adapt it for downstream tasks with new reward functions.

A remarkable aspect of GFlowNets is the ability to demonstrate the adaptability in generating a task-specific policy. By utilizing the pre-trained OC-GFN model with outcome-conditioned flows $F(s|y)$ and policies $P_F(s'|s, y)$, we can directly obtain a policy that samples according to a new task-specific reward function $R(s) = r(y)$ according to Eq. (6), which is based on (Bengio et al., 2023). A detailed analysis for this can be found in Appendix B.2.

$$P_F^r(s'|s) = \frac{\sum_y r(y)F(s|y)P_F(s'|s, y)}{\sum_y r(y)F(s|y)} \tag{6}$$

Eq. (6) serves as the foundation for converting a pre-trained OC-GFN model to handle downstream tasks with new and even out-of-distribution rewards. Intriguingly, *this conversion can be achieved without any re-training* for OC-GFN on downstream tasks. This sets it apart from the typical fine-tuning process in reinforcement learning, which typically requires re-training to adapt a policy, since they generally learn reward-maximizing policies that may discard valuable information.

Directly estimating the above summation often necessitates the use of Monte-Carlo averaging for making each decision. Yet, this can be computationally expensive in high-dimensional outcome spaces if we need to calculate this marginalization at each decision-making step, which leads to slow thinking. To improve its efficiency in complex scenarios, it is essential to develop strategies that are both efficient while maintaining accurate estimations.

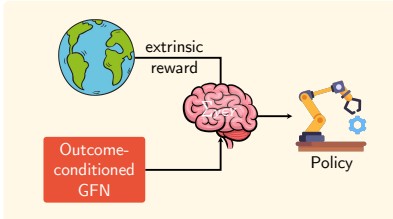 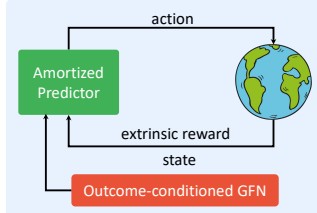

Figure 2: *Left*: Convert the outcome-conditioned GFlowNet to downstream tasks without re-training the networks. *Right*: An efficient amortized predictor in the supervised fine-tuning phase.

**Learning an Amortized Predictor**    In this section, we propose a novel approach to approximate this marginal by learning an amortized predictor.

Concretely, we propose to estimate the intractable sums in the numerator in Eq. (6) by a numerator network $N(s'|s) \approx \sum_y r(y)F(s|y)P_F(s'|s,y)$. This would allow us to efficiently estimate the intractable sum with the help of $N(s'|s)$ that directly estimates $N(\cdot|s)$ for any state $s$, which could benefit from the generalizable structure of outcomes with neural networks. We can also obtain the corresponding policy by $P_F^r(s'|s) = N(s'|s)/\sum_{s'} N(s'|s)$. For learning the numerator network $N(\cdot|s)$, we need to have a sampling policy for sampling outcomes $y$ given states $s$ and next states $s'$, which can be achieved by

$$Q(y|s',s) = \frac{r(y)F(s|y)P_F(s'|s,y)}{N(s'|s)}. \tag{7}$$

Therefore, we have the following constraint according to Eq. (7)

$$N(s'|s)Q(y|s',s) = r(y)F(s|y)P_F(s'|s,y), \tag{8}$$

based on which we derive the corresponding loss function by minimizing their difference by training in the log domain, i.e.,

$$\mathcal{L}_{amortized} = \left(\log N(s'|s) + \log Q(y|s',s) - \log r(y) - \log F(s|y) - \log P_F(s'|s,y)\right)^2. \tag{9}$$

**Theoretical justification.**    In Proposition 4.2, we show that $N(s'|s)$ can correctly estimate the summation when it is trained to completion and the distribution of outcomes $y$ has full support.

**Proposition 4.2.** *Suppose that $\forall(s, s', y)$, $\mathcal{L}_{amortized}(s, s', y) = 0$, then the amortized predictor $N(s'|s)$ estimates $\sum_y r(y)F(s|y)P_F(s'|s,y)$.*

The proof can be found in Appendix B.3. Proposition 4.2 justifies the use of the numerator network as an efficient alternative for estimating the computationally intractable summation in Eq. (6).

**Empirical validation.**    We now investigate the converted/learned sampling policy in the standard GridWorld domain (Bengio et al., 2021), which has a multi-modal reward function. More details about the setup is in Appendix C.1 due to space limitation. We visualize the last $2 \times 10^5$ samples from different baselines including training GFN from scratch, OC-GFN with the Monte Carlo-based estimation with Eq. (6) and the amortized marginalization approach with Eq. (9).

As shown in Figure 3(b), directly training GFN with trajectory balance (Malkin et al., 2022) can suffer from the mode collapse problem and fail to discover all modes of the target distribution in Figure 3(a). On the contrary, we can directly obtain a policy from the pre-trained OC-GFN model that samples proportionally to the target rewards as shown in Figure 3(c), while fine-tuning OC-GFN with the amortized inference could also match the target distribution as in Figure 3(d), which validates its effectiveness in estimating the marginal.

**Practical implementation.**    In practice, we can train the amortized predictor $N(\cdot|s)$ and the sampling policy $Q(\cdot|s',s)$ in a GFlowNet-like procedure. We sample $(s, s')$ with $N$ by interacting with the environment. This can also be realized by sampling from the large and diverse dataset $\mathcal{D}$ obtained in the pre-training stage. We then sample outcomes $y$ from the sampling policy $Q(\cdot|s',s)$, which can incorporate its tempered version or with $\epsilon$-greedy exploration (Bengio et al., 2021) for obtaining rich distributions of $y$. We can then learn the amortized predictor and the sampling policy by

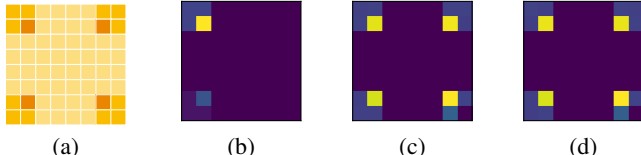

|       (a)       |       (b)       |       (c)       |       (d)       |

Figure 3: Distribution of $2 \times 10^5$ samples from different baselines. (a) Target distribution. (b) GFN (from scratch). (c) OC-GFN (Monte Carlo-based). (d) OC-GFN (Amortizer-based).

one of these sampled experiences. Finally, we can derive the policy that will be converted to downstream tasks by $\hat{P}_F^r(s'|s) = N(s'|s)/\sum_{s'} N(s'|s)$. The resulting training algorithm is summarized in Algorithm 2 and the right part in Figure 2.

## 5 EXPERIMENTS

In this section, we conduct extensive experiments to better understand the effectiveness of our approach and aim to answer the following key questions: **(i)** How do outcome-conditioned GFlowNets (OC-GFN) perform in the reward-free unsupervised pre-training stage? **(ii)** What is the effect of key modules? **(iii)** Can OC-GFN transfer to downstream tasks efficiently? **(iv)** Can they scale to complex and practical scenarios like biological sequence design?

### 5.1 GRIDWORLD

We first conduct a series of experiments on GridWorld (Bengio et al., 2021) to understand the effectiveness of the proposed approach. In the reward-free unsupervised pre-training phase, we train a (unconditional) GAFN (Pan et al., 2023b) and an outcome-conditioned GFN (OC-GFN) on a map without task-specific rewards, where GAFN is trained purely from self-supervised intrinsic rewards according to Algorithm 1. We investigate how well OC-GFN learns in the unsupervised pre-training stage, while the supervised fine-tuning stage has been discussed in Section 4.2. Each algorithm is run for 3 different seeds and the mean and standard deviation are reported. A detailed description of the setup and hyperparameters can be found in Appendix C.1.

**Outcome distribution.** We first evaluate the quality of the exploratory data collected by GAFN, as it is essential for training OC-GFN with a rich distribution of outcomes. We demonstrate the sample distribution from GAFN in Figure 4(a), which has great diversity and coverage, and validates its effectiveness in collecting unlabeled exploratory trajectories for providing diverse target outcomes for OC-GFN to learn from.

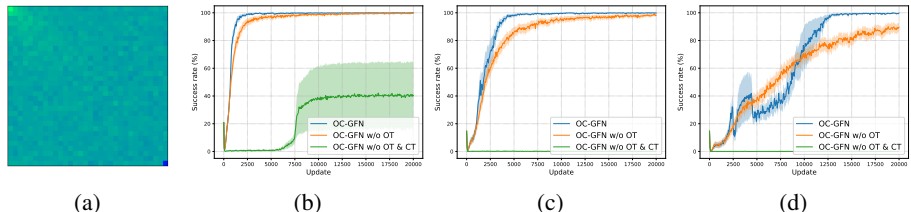

|       (a)       |       (b)       |       (c)       |       (d)       |

Figure 4: Results in GridWorld in different with different scales of the task. (a) Outcome distribution. (b)-(d) Success rate of OC-GFN and its variants in small, medium, and large maps, respectively.

**Outcome reaching performance.** We then investigate the key designs in pre-training OC-GFN by analyzing its success rate for achieving target outcomes. We also ablate key components to investigate the effect of contrastive training (CT) and outcome teleportation (OT). The success rates of OC-GFN and its variants are summarized in Figures 4(b)-(d) in different sizes of the map from small to large. As shown, OC-GFN can successfully learn to reach specified outcomes with a success rate of nearly $100\%$ at the end of training in maps with different sizes. Disabling the outcome teleportation component (OC-GFN w/o OT) leads to lower sample efficiency, in particular in larger maps. Further deactivating the contrastive training process (OC-GFN w/o OT & CT) fails to learn well as the size of the outcome space grows, since the agent can hardly collect successful trajectories for meaningful updates. This indicates that both contrastive training and outcome teleportation are important for successfully training OC-GFN in large outcome spaces. It is also worth noting that

outcome teleportation can significantly boost learning efficiency in problems with a combinatorial explosion of outcome spaces (e.g., sequence generation in the following sections).

**Outcome-conditioned behaviors.** We visualize the learned behaviors of OC-GFN in Figure 5, where the red and green squares denote the starting point and the goal, respectively. We find that OC-GFN can not only reach different outcomes with a high success rate, but also discover diverse trajectories leading to a specified outcome $y$ with the forward policy $P_F$. OC-GFN is able to generate diverse trajectories, which is different from typical goal-conditioned RL approaches (Schaul et al., 2015) that usually only learn a single solution to the goal state. This ability can be very helpful for generalizing to downstream tasks with similar structure but subtle changes (e.g., obstacles in the maze) (Kumar et al., 2020).

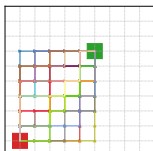

Figure 5: Behaviors of OC-GFN.

**Applicability to other objectives.** We now investigate the versatility of our approach by building it upon another recent GFlowNets learning objective with SubTB (Madan et al., 2022). We investigate the outcome-reaching performance of OC-SubTB in the pre-training stage with a medium map as in Figure 6(a), while the learned sample distribution by fine-tuning the outcome-conditioned flows and policies of OC-SubTB with our amortized predictor is shown in Figure 6(b). As shown, OC-SubTB can successfully learn to reach target outcomes, where the proposed contrastive training and outcome teleportation method achieve consistent gains in efficiency. Fine-tuning OC-SubTB can also discover all the modes, as opposed to training a GFN (Malkin et al., 2022) from scratch (Figure 3). A more detailed discussion is in Appendix B.2.

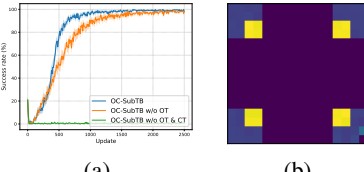

(a)          (b)

Figure 6: OC-SubTB in GridWorld.

## 5.2 BIT SEQUENCE GENERATION

We study the bit sequence generation task (Malkin et al., 2022), where the agent generates sequences of length $n$. We follow the same procedure for pre-training OC-GFN without task-specific rewards as in Section 5.1 with more details in Appendix C.1.

**Analysis of the unsupervised pre-training stage.** We first analyze how well OC-GFN learns in the unsupervised pre-training stage by investigating its success rate for achieving specified outcomes in bit sequence generation problems with different scales including small, medium, and large. As shown in Figures 7(a)-(c), OC-GFN can successfully reach outcomes, leading to a high success rate as summarized in Figure 7(d). It is also worth noting that it fails to learn when the outcome teleportation module is disabled due to the particularly large outcome spaces. In such highly challenging scenarios, the contrastive training paradigm alone does not lead to efficient learning.

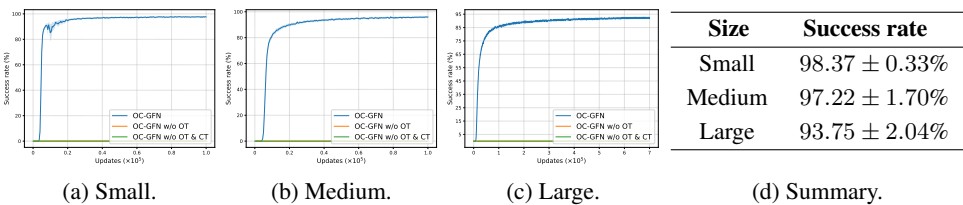

| (a) Small. | (b) Medium. | (c) Large. | (d) Summary. |

Figure 7: Success rates in the bit sequence generation task with different scales of the task.

**Analysis of the Supervised Fine-Tuning Stage.** After we justified the effectiveness of training OC-GFN in the unsupervised pre-training stage, we study whether fine-tuning the model with the amortized approach can enable faster mode discovery when adapting to downstream tasks (as described in Appendix C.1). We compare the proposed approach against strong baselines including training a GFN from scratch (Bengio et al., 2023), Metropolis-Hastings-MCMC (Dai et al., 2020), and Deep Q-Networks (DQN) (Mnih et al., 2015). We evaluate each method in terms of the number of modes discovered during the course of training as in previous works (Malkin et al., 2022). We summarize the normalized (between 0 and 1 to facilitate comparison across tasks) number of modes averaged over each downstream task in Figure 8, while results for each individual downstream task can be found in Appendix C.5 due to space limitation.

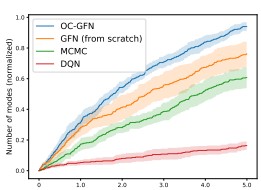

Figure 8: Results in bit sequence generation.

We find that OC-GFN significantly outperforms baselines in mode discovery in both efficiency and performance, while DQN gets stuck and does not discover diverse modes due to the reward-maximizing nature, and MCMC fails to perform well in problems with larger state spaces.

## 5.3 TF Bind Generation

We study a more practical task of generating DNA sequences with high binding activity with targeted transcription factors (Jain et al., 2022). We consider 30 different downstream tasks studied in (Barrera et al., 2016), which conducted biological experiments to determine the binding properties between a range of transcription factors and every conceivable DNA sequence.

**Analysis of the unsupervised pre-training stage.** We analyze how well OC-GFN learns by evaluating its success rate. Figure 9 shows that OC-GFN can successfully achieve the targeted goals with a success rate of nearly 100% in the practical task of generating DNA sequences.

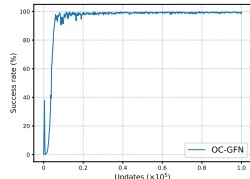

**Analysis of the supervised fine-tuning stage.** We investigate how well OC-GFN can transfer to downstream tasks for generating TF Bind 8 sequences. We tune the hyperparameters on the task PAX3_R270C_R1, and evaluate the baselines on the other 29 downstream tasks. We investigate the learning efficiency and performance in terms of the number of modes and the top-$K$ scores during the course of training. The results in 2 downstream tasks are shown in Figure 10, with the full results in Appendix C.6 due to space limitation. Figures 10(a)-(b) illustrate the number of discovered modes while Figures 10(c)-(d) shows the top-$K$ scores. We

Figure 9: Success rate in TF Bind generation.

find that the proposed approach discovers more diverse modes faster and achieves higher top-$K$ ($K = 100$) scores compared to baselines. We summarize the rank of each baseline in all 30 downstream tasks (Zhang et al., 2021) in Appendix C.6, where OC-GFN ranks highest compared with other methods. We further visualize in Figure 10(e) the t-SNE plots of the TF Bind 8 sequences discovered by transferring OC-GFN with the amortized approach and the more expensive but competitive training of a GFN from scratch. As shown, training a GFN from scratch only focuses on limited regions while the OC-GFN has a greater coverage.

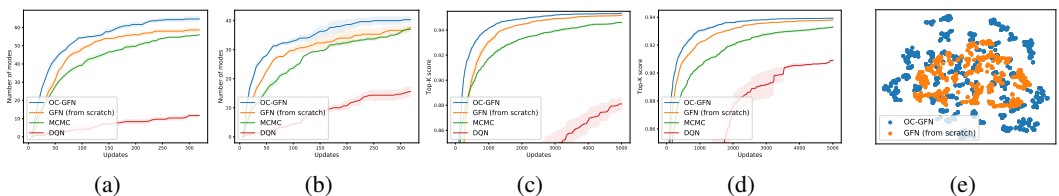

(a)  (b)  (c)  (d)  (e)

Figure 10: Results in the TF Bind sequence generation task in different downstream tasks.

## 5.4 RNA Generation

We now study a larger task of generating RNA sequences that bind to a given target introduced in (Lorenz et al., 2011). We follow the same procedure as in Section 5.3, where details are in Appendix C.1. We consider four different downstream tasks from ViennaRNA (Lorenz et al., 2011), each considering the binding energy with a different target as a reward. The left part in Figure 11 shows that OC-GFN achieves a success rate of almost 100% at the end of the pre-

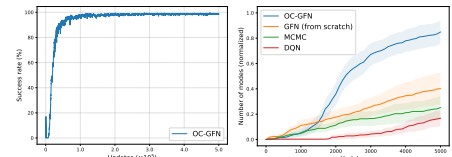

Figure 11: Results in RNA generation. *Left*: success rate. *Right*: number of modes (normalized).

training stage. The right part in Figure 11 summarizes the averaged normalized (between 0 and 1 to facilitate comparison across tasks) performance in terms of the number of modes averaged over four downstream tasks, where the performance for each individual task is shown in Appendix C.7. We observe that OC-GFN is able to achieve much higher diversity than baselines, indicating that the pre-training phase enables the OC-GFN to explore the state space much more efficiently.

## 5.5 Antimicrobial Peptide Generation

To demonstrate the scalability of our approach to even more challenging and complex scenarios, we also evaluate it on the biological task of generating antimicrobial peptides (AMP) with lengths 50 (Jain et al., 2022) (resulting in an outcome space of $20^{50}$ candidates). Figure 12(a) demonstrates

the success rate of OC-GFN for achieving target outcomes, which can still reach a high success rate in particularly large outcome spaces with our novel training paradigm according to contrastive learning and fast goal propagation. Figure 12(b) shows the number of modes discovered, where OC-GFN provides consistent improvements, which validates the efficacy of OC-GFN to successfully scale to much larger-scale tasks.

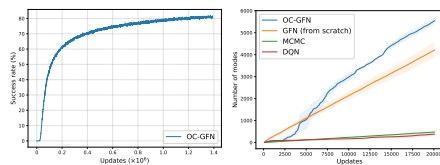

(a) Success rate.   (b) Number of modes.
Figure 12: Results in AMP generation.

## 6 CONCLUSION

We propose a novel method for unsupervised pre-training of GFlowNets through an outcome-conditioned GFlowNet, coupled with a new approach to efficiently fine-tune the pre-trained model for downstream tasks. Our work opens the door for GFlowNets to be pre-trained for fine-tuning for downstream tasks. Empirical results on the standard GridWorld domain validate the effectiveness of the proposed approach in successfully achieving targeted outcomes and the efficiency of the amortized predictor. We also conduct extensive experiments in the more complex and challenging biological sequence design tasks to demonstrate its practical scalability. Our method greatly improves learning performance compared with strong baselines including training a GFlowNet from scratch, particularly in tasks which are challenging for GFlowNets to learn.

## ACKNOWLEDGMENTS

The authors acknowledge funding from CIFAR, Genentech, Samsung, and IBM.

## ETHICS STATEMENT

We do not foresee any immediate negative societal impact of our work. Our work is motivated by the need for ways to accelerate scientific problems. However, we do note that there is a potential risk of dual use of the technology by nefarious actors (Urbina et al., 2022), as with all work around this topic.

## REPRODUCIBILITY

All details of our experiments are discussed in Appendix C with a detailed description of the task, network architectures, hyper-parameters for baselines, and setup. We implement all baselines and environments based on open-source repositories described in Appendix C. The proofs of propositions can be found in Appendix B. Limitations are discussed in Appendix D.

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

## A   ADDITIONAL DETAILS OF GFLOWNETS

### A.1   DETAILS FOR THE TRAINING LOSSES

As introduced in Section 2, the flow consistency constraint for detailed balance (DB) (Bengio et al., 2023) is $F(s)P_F(s'|s) = F(s')P_B(s|s')$ for $s \rightarrow s' \in \mathcal{A}$. In practice, we train DB in the log-scale for stability (Bengio et al., 2021), i.e.,

$$\forall s \rightarrow s' \in \mathcal{A}, \quad \mathcal{L}_{\text{DB}}(s, s') = \left( \log \frac{F(s)P_F(s'|s)}{F(s')P_B(s|s')} \right)^2, \tag{10}$$

and terminal flows $F(x)$ to match the corresponding rewards $R(x)$. The learnable objects here are the state flow $F(s)$, forward policy $P_F(s'|s)$, and backward policy $P_B(s|s')$, which are parameterized by neural networks. For GAFlowNets (Pan et al., 2023b) that consider intrinsic intermediate rewards $r_i(s \rightarrow s')$ into the flow network, we also train the model in log-scale according to Eq. (2), i.e.,

$$\forall s \rightarrow s' \in \mathcal{A}, \quad \mathcal{L}_{\text{GAFN (DB)}}(s, s') = \left( \log \frac{F(s)P_F(s'|s)}{F(s')P_B(s|s') + r_i(s \rightarrow s')} \right)^2. \tag{11}$$

The learnable objects are the same as in DB, except that $r_i(s \rightarrow s')$ is also learnable, which is represented by intrinsic motivation by random network distillation (Burda et al., 2018)).

## B   PROOFS

### B.1   PROOF OF PROPOSITION 4.1

**Proposition 4.1** *If $\mathcal{L}_{\text{OC-GFN}}(\tau, y) = 0$ for all trajectories $\tau$ and outcomes $y$, then the outcome-conditioned forward policy $P_F(s'|s, y)$ can successfully reach any target outcome $y$.*

*Proof.* As $\mathcal{L}_{\text{OC-GFN}}(\tau, y) = 0$ is satisfied for all trajectories $\tau = \{s_0, \cdots, s_n = x\}$ and outcomes $y$, we have that

$$F(s_0|y) \prod_{t=0}^{n-1} P_F(s_{t+1}|s_t, y) = \prod_{t=0}^{n-1} R(x|y)P_B(s_t|s_{t+1}, y). \tag{12}$$

Since $R(x|y)$ is either 1 or 0 in outcome-condition tasks, we get that

$$F(s_0|y) \prod_{t=0}^{n-1} P_F(s_{t+1}|s_t, y) = R(x|y) \prod_{t=0}^{n-1} P_B(s_t|s_{t+1}, y). \tag{13}$$

Then, the probability of reaching the target outcome $y$ is

$$P(x = y|y) = \sum_{\tau, s_n = y} P(\tau|y). \tag{14}$$

By definition, we have that

$$P(\tau|y) = \prod_{t=0}^{n-1} P_F(s_{t+1}|s_t, y). \tag{15}$$

Therefore, we get that

$$P(x = y|y) = \sum_{\tau, s_n = y} \prod_{t=0}^{n-1} P_F(s_{t+1}|s_t, y). \tag{16}$$

Combining Eq. (13) with Eq. (16), and due to the law of total probability, we obtain that

$$\begin{aligned} F(s_0|y)P(x = y|y) &= \sum_{\tau, s_n = y} R(x|y) \prod_{t=0}^{n-1} P_B(s_t|s_{t+1}, y) \\ &= R(y|y) \sum_{\tau, s_n = y} \prod_{t=0}^{n-1} P_B(s_t|s_{t+1}, y) \\ &= \sum_{\tau, s_n = y} \prod_{t=0}^{n-1} P_B(s_t|s_{t+1}, y) = 1, \end{aligned} \tag{17}$$

where we note that $R(y|y) = 1$ according to the definition of the reward function (which is binary).

With the same analysis for the case where the agent fails to reach the target outcome $y$, i.e., $x \neq y$ and $R(x|y) = 0$, we have that

$$\forall x \neq y, \quad F(s_0|y)P(x|y) = 0. \tag{18}$$

Combing Eq. (17) with Eq. (18), we have that $P(x = y|y) = 1$, i.e., the outcome-conditioned forward policy $P_F(s'|s, y)$ can successfully reach any target outcome $y$.

$\square$

## B.2 ANALYSIS OF THE CONVERSION POLICY

We now elaborate on more details about the effect of Eq. (6) in the text based on (Bengio et al., 2023).

When the outcome-conditioned GFlowNet (OC-GFN) is trained to completion, the following flow consistency constraint in the edge level is satisfied for intermediate states.

$$F(s|y)P_F(s'|s, y) = F(s'|y)P_B(s|s', y) \tag{19}$$

We define the state flow function as $F^r(s) = \sum_y r(y)F(s|y)$, and the backward policy as

$$P_B^r(s|s') = \frac{\sum_y r(y)F(s'|y)P_B(s|s', y)}{\sum_y r(y)F(s'|y)}, \tag{20}$$

while the forward policy is defined in Eq. (6), i.e.,

$$P_F^r(s'|s) = \frac{\sum_y r(y)F(s|y)P_F(s'|s, y)}{\sum_y r(y)F(s|y)} \tag{21}$$

Then, we have

$$F^r(s)P_F^r(s'|s) = \sum_y r(y)F(s|y)P_F(s'|s, y), \tag{22}$$

and

$$F^r(s')P_B^r(s|s') = \sum_y r(y)F(s'|y)P_B(s|s', y). \tag{23}$$

Combining the above equations, we have that

$$F^r(s)P_F^r(s'|s) = F(s')^r P_B^r(s|s'), \tag{24}$$

which corresponds to a new flow consistency constraint in the edge level. A more detailed proof can be found in (Bengio et al., 2023).

**Discussion about applicability**   It is also worth noting that OC-GFN should be built upon the detailed balance objective (Bengio et al., 2023) discussed in Section 2 or other variants which learn flows. Instead, the trajectory balance objective (Malkin et al., 2022), does not learn a state flow function necessary for converting the pre-trained OC-GFN model to the new policy $\pi_r$.

We obtain the learning objective for OC-GFN when built upon sub-trajectory balance (SubTB) (Madan et al., 2022) for a sub-trajectory $\tau_{i:j} = \{s_i, \cdots, s_j\}$ as in Eq. (25) following Section 4.1.1.

$$F(s_i|y)\prod_{t=i}^{j-1} P_F(s_{t+1}|s_t, y) = F(s_j|y)\prod_{t=i}^{j-1} P_B(s_t|s_{t+1}, y). \tag{25}$$

Further incorporating our proposed outcome teleportation technique for OC-GFN (SubTB) results in the training objective as in Eq. (26), which is trained in the log domain:

$$\mathcal{L}_{\text{OC-GFN (SubTB)}}(\tau) = \sum_{\tau_{i:j} \in \tau} w_{ij} \left( \log F(s_i|y) + \sum_{t=i}^{j-1} \log P_F(s_{t+1}|s_t, y) \right.$$
$$\left. - \log F(s_j|y) - \sum_{t=i}^{j-1} \log P_B(s_t|s_{t+1}, y) - \log R(x|y) \right)^2, \tag{26}$$

where $w_{ij} = \frac{\lambda^{j-i}}{\sum_{0 \le i < j \le n} \lambda^{j-i}}$ denotes the weight for the sub-trajectory $\tau_{i:j}$, and $\lambda$ is a hyperparameter for controlling the weights. Besides the performance of OC-SubTB studied in the GridWorld task in Section 5.1, we also demonstrate its effectiveness in two TF Bind generation tasks by studying the success rate of OC-SubTB in the unsupervised pre-training stage, and the number of modes and top-$K$ scores for fine-tuning OC-SubTB. Results are summarized in Figure 13, where we observe consistently high success rate (nearly $100\%$ at the end of the pre-training stage) and improved performance in the fine-tuning stage.

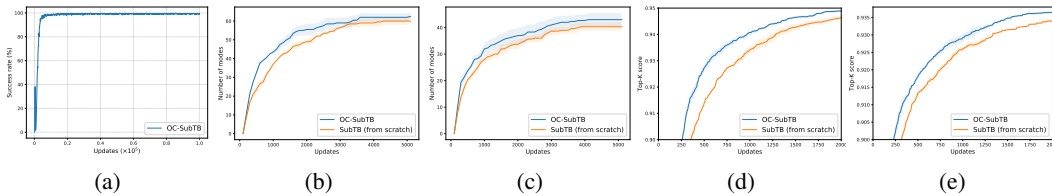

(a)      (b)      (c)      (d)      (e)

Figure 13: Results of OC-GFN when built upon SubTb in the TF Bind sequence generation task in different downstream tasks (PAX3_R270C_R1 and HOXB7_M190L_R1). (a) Success rate of pre-training OC-SubTB. (b)-(c) The number of modes discovered. (d)-(e) Top-$K$ scores.

### B.3 PROOF OF PROPOSITION 4.2

**Proposition 4.2** *Suppose that $\forall (s, s', y)$, $\mathcal{L}_{amortized}(s, s', y) = 0$, then the amortized predictor $N(s'|s)$ estimates $\sum_y r(y) F(s|y) P_F(s'|s, y)$.*

*Proof.* As $\mathcal{L}_{\text{amortized}}(s, s', y) = 0$ for all states $s$, next states $s'$, and outcomes $y$, we have that

$$\forall s, s', y, \quad N(s'|s) Q(y|s', s) = r(y) F(s|y) P_F(s'|s, y). \tag{27}$$

Therefore, by summing over all possible outcomes $y$ on both sides, we obtain that

$$\forall s, s', \quad \sum_y N(s'|s) Q(y|s', s) = \sum_y r(y) F(s|y) P_F(s'|s, y). \tag{28}$$

As $\sum_y Q(y|s', s) = 1$, we get that

$$\forall s, s', \quad N(s'|s) = \sum_y r(y) F(s|y) P_F(s'|s, y). \tag{29}$$

Based on the above analysis, we get that the amortized predictor $N(s'|s)$ estimates the marginal $\sum_y r(y) F(s|y) P_F(s'|s, y)$.

$\square$

## C EXPERIMENTAL DETAILS

All baseline methods are implemented based on the open-source implementation,[1] where we follow the default hyperparameters and setup as in (Bengio et al., 2021). The code will be released upon publication of the paper.

### C.1 EXPERIMENTAL SETUP

**GridWorld** We employ the same standard reward function for GridWorld from (Bengio et al., 2021) as in Eq. (30), where the target reward distribution is shown in Figure 3(a) with 4 modes located near the corners of the maze.

$$R(x) = \frac{1}{2} \prod_i \mathbb{I}(0.25 < |x_i/H - 0.5|) + 2 \prod_i \mathbb{I}(0.3 < |x_i/H - 0.5| < 0.4) + 10^{-6}. \tag{30}$$

---

[1] https://github.com/GFNOrg/gflownet

The GFlowNet model is a feedforward network consisting of two hidden layers with 256 hidden units per layer using LeakyReLU activation. We use a same network structure for the outcome-conditioned GFlowNet model, except that we concatenate the state and outcome as input to the network. For the unconditional GAFlowNet model, we follow (Pan et al., 2023b) and leverage random network distillation as the intrinsic reward mechanism. The coefficient for intrinsic rewards is set to be 1.0 (which purely learns from intrinsic motivation without task-specific rewards). We also use a same network structure for the amortized predictor network. We train all models with the Adam (Kingma & Ba, 2015) optimizer (learning rate is 0.001) based on samples from a parallel of 16 rollouts in the environment.

**Bit Sequence Generation** We consider generating bit sequences with fixed lengths based on (Malkin et al., 2022). The GFlowNet model is a feedforward network that consists of 2 hidden layers with 2048 hidden units with ReLU activation. The exploration strategy is $\epsilon$-greedy with $\epsilon = 0.0005$, while we set the sampling temperature to 1.0, and use a reward exponent of 3. The learning rate for training the GFlowNet model is $5 \times 10^{-3}$ with the Adam optimizer, with a batch size of 16. We use a same network structure for the outcome-conditioned GFlowNet model and the amortized predictor network. We train all models for 50000 iterations, using a parallel of 16 rollouts in the environment.

**TF Bind and RNA Generation** For the TFBind-8 generation task, we follow the same setup as in (Jain et al., 2022). The vocabulary consists of 4 nucleobases, and the trajectory length is 8 and 14. The GFlowNet model is a feedforward network that consists of 2 hidden layers with 2048 hidden units and ReLU activation. The exploration strategy is $\epsilon$-greedy with $\epsilon = 0.001$, while the reward exponent is 3. The learning rate for training the GFlowNet model is $1e-4$, with a batch size of 32. We train all models for 5000 iterations.

**AMP Generation** We basically follow the same setup for the antimicrobial peptide generation task as in Malkin et al. (2022); Jain et al. (2022), where we consider generating AMP sequence with length 50. The GFlowNet model is an MLP that consists of 2048 hidden layers with 2 hidden units with ReLU activation. The exploration strategy is $\epsilon$-greedy with $\epsilon = 0.01$, while the sampling temperature is set to 1, and uses a reward exponent of 3. The lr for training the GFlowNet model is 0.001, and the batch size is 16.

**Metrics** We use the same evaluation metrics as in the literature in GFlowNets (Bengio et al., 2021; Malkin et al., 2022; Madan et al., 2022; Pan et al., 2023b;a). The number of modes is calculated using a sphere-exclusion procedure. Specifically, the generated candidates are sorted by reward, and then scanning down this sorted list, a candidate is added to the list of modes if it is above a certain reward threshold and is further away than a distance threshold $\delta$ from all other modes. For the normalized number of modes, we normalize the number of modes discovered in each task by min-max normalization, and average this value across different tasks.

## C.2 ADDITIONAL ABLATION STUDY OF OC-GFN IN THE UNSUPERVISED PRE-TRAINING STAGE

We now further investigate the impact of the GAFN (Pan et al., 2023b) in the unsupervised pre-training stage as discussed in Section 4.1.1, based on the learning objective for OC-GFN presented in Eq. (3). We investigate the influence of GAFN by ablating the model design and replacing it with a random agent, and a GFlowNet agent that is trained with task-irrelevant rewards (i.e., $R(x) = 1$) in the unsupervised pre-training stage, while the GAFN is learned by pure intrinsic motivation with random network distillation (Burda et al., 2018). We conduct this ablation study based on the GridWorld environment as in Section 5.1. Please note that the experiments are run for about 2500 updates only due to time limitation in the rebuttal period (and therefore the success rate is not around 100% yet). It is also worth noting that the effect of completely removing the GAFN part has been studied in Section 5.1, where OC-GFN w/ OT & CT in Figure 4 in the text examines the significance of the contrastive training technique.

We compare the success rate of each method for reaching given outcomes following Section 5.1. As shown in Figure 14, OC-GFN (GAFN) learns much more efficiently than the other two variants, where the GAFN model is replaced with a GFN model trained with task-irrelevant rewards (OC-GFN (GFN)) or a random agent (OC-GFN (random)), with a more significant margin in larger maps.

We also study the generated outcome distribution by ablating this model design to gain a better understanding during the learning process. Figure 15 demonstrates the outcome distribution generated

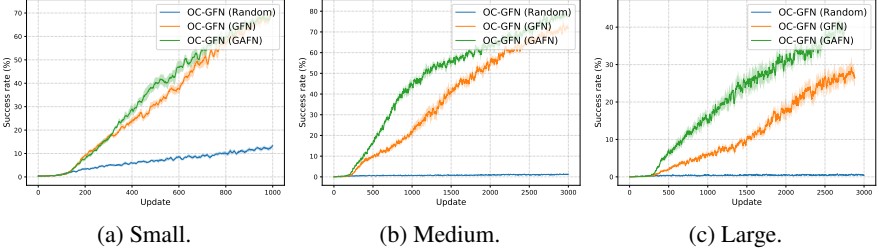

(a) Small.        (b) Medium.        (c) Large.

Figure 14: Ablation study of OC-GFN in the unsupervised pre-training stage in GridWorld in success rate.

by each method at the same iteration (with 2500 updates) in GridWorld with a large map. As shown, OC-GFN (GAFN) motivates the agent to explore unfamiliar regions, resulting in a more diverse outcome distribution for training the OC-GFN and improving its learning efficiency. This observation is consistent with Figure 14, demonstrating the importance of the GAFN model.

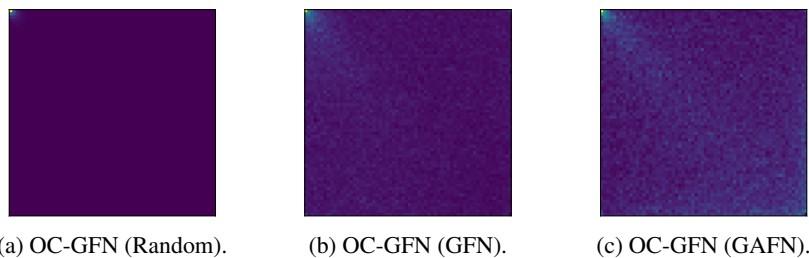

(a) OC-GFN (Random).   (b) OC-GFN (GFN).   (c) OC-GFN (GAFN).

Figure 15: Outcome distribution for variants of OC-GFN in the unsupervised pre-training stage in GridWorld with a large map.

## C.3 Additional Results of a Direct Pre-Training Scheme

One can also pre-train a GFlowNet on task A and then fine-tune the model on task B. However, this simple scheme for pre-training GFlowNets does not lead to a universally applicable strategy for efficient transfer. This is because the GFlowNet learns to sample proportionally from the reward function of task A during the pre-training stage, and thus, if there is not enough shared structure between the rewards for task A and task B (or the goals are unrelated), the GFlowNet model pre-trained on task A may not be beneficial for the fine-tuning stage on task B, and could potentially even have a negative impact.

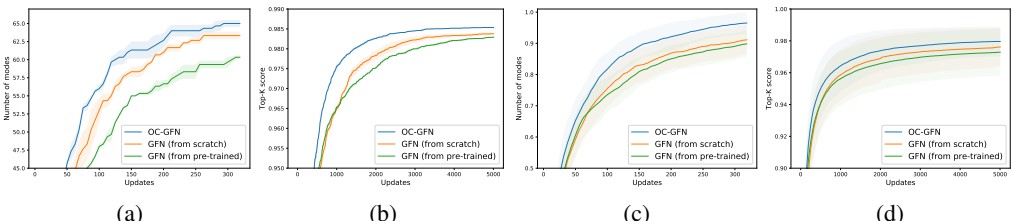

(a)      (b)      (c)      (d)

Figure 16: Comparison results of GFN (from pre-trained) in TF Bind generation tasks. (a)-(b) Mode discovery and top-$K$ scores in SIX6_T165A_R1. (c)-(d) Normalized mode discovery and normalized top-$K$ scores averaged over 10 downstream tasks.

We now compare our proposed OC-GFN with fine-tuning this schema (which is pre-trained on PAX3_R270C_R1) on 10 TF Bind generation tasks (as described in Section 5.3). Figures 16(a)-(b) demonstrate the results in terms of mode discovery and top-$K$ scores on one downstream task (SIX6_T165A_R1) for demonstration, while Figures 16 summarize the normalized mode discovery and top-$K$ performance averaged over each downstream task. As shown, this GFN (from pre-trained) schema does not provide universal effectiveness (which can lead to lower efficiency in mode discovery and top-$K$ performance), as it is sensitive to the reward structures of the downstream

tasks. It has also been highlighted in previous works (Bengio et al., 2021) that GFlowNets are prone to overfitting on certain tasks, which could further result in worse performance compared to training a GFlowNet from scratch on task B. On the contrary, we cast the problem of unsupervised pre-training of GFlowNet with self-supervised learning, which *learns to reach any outcome* for learning a *functional understanding* of the task. Therefore, it provides a universally effective strategy to different downstream tasks, and is agnostic to the reward structures.

### C.4 ADDITIONAL RESULTS OF COMPARISON WITH RL METHODS

We include an additional comparison of OC-GFN with other two strong RL methods including PPO (Schulman et al., 2017) and SAC (Haarnoja et al., 2018) with automatic tuning of the entropy coefficient, in the same two downstream tasks of TF Bind generation as in Figure 10 in Section 5.3 (PAX3_R270C_R1 and HOXB7_M190L_R1). Figure 17 demonstrates the comparison results in terms of the number of discovered modes ((a)-(b)) during learning and the top-$K$ scores ((c)-(d)). As shown, the reward-maximization nature of RL approaches limits the resulting solution diversity (Bengio et al., 2021; 2023), which underperforms GFN (from scratch) in the number of discovered modes and top-$K$ scores, and OC-GFN consistently outperforms baselines in the mode-seeking ability.

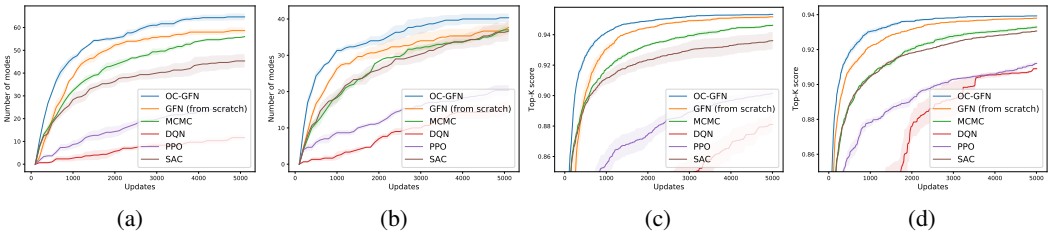

Figure 17: Additional comparison results with other RL methods (PPO and SAC) in different downstream tasks in TF Bind generation. (a)-(b): Number of modes. (c)-(d): Top-$K$ scores.

**Computation cost** We now compare the computation cost of each baseline method in Figure 17 in terms of memory and time (for collecting experiences and for training the model). As shown in Figure 18, training OC-GFN only leads to marginal computation overhead.

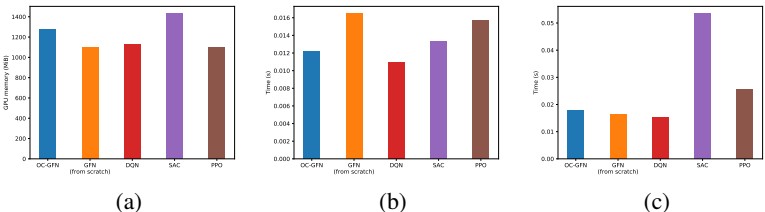

Figure 18: Comparison of computation costs. (a) GPU Memory. (b) Collect experiences. (c) Train models.

### C.5 ADDITIONAL RESULTS ON THE BIT SEQUENCE GENERATION TASK

As shown in Figures 19(a)-(c), OC-GFN is more efficient at discovering modes and discovers more modes, compared with baselines in different scales of the task, from small to large. DQN gets stuck and does not discover diverse modes due to the reward-maximizing nature of regular reinforcement learning algorithms, while MCMC fails to perform well in problems with larger state spaces. Besides different scales of the problem, we also consider different downstream tasks as in Figures 19(d)-(e), which further demonstrate the effectiveness of OC-GFN in the supervised fine-tuning stage.

### C.6 ADDITIONAL RESULTS ON THE TF BIND GENERATION TASK

The full results in the TF Bind generation task are demonstrated in Figure 21 for the 30 downstream from (Lorenz et al., 2011). The mean rank averaged over all tasks is summarized in Table 1, where OC-GFN significantly outperforms baselines and is able to discover more modes in a more efficient way.

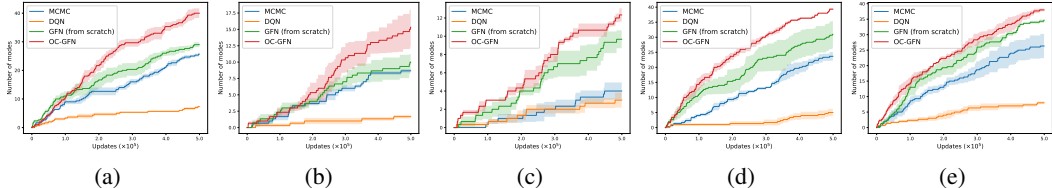

| | (a) | (b) | (c) | (d) | (e) |

Figure 19: Results in the bit sequence generation task in different with different scales of the task.

Table 1: Mean rank of each baseline in all downstream tasks (the lower the better).

| | DQN | MCMC | GFN (from scratch) | OC-GFN |
|---|---|---|---|---|
| Number of modes | 4.00 | 2.77 | 1.97 | **1.27** |
| Top-$K$ reward | 4.00 | 2.80 | 2.00 | **1.20** |

### C.7 ADDITIONAL RESULTS ON THE RNA GENERATION TASK

The full results in the RNA sequence generation task are demonstrated in Figure 20 for the four downstream from (Lorenz et al., 2011), where OC-GFN significantly outperforms baseline methods achieving better diversity in a more efficient manner.

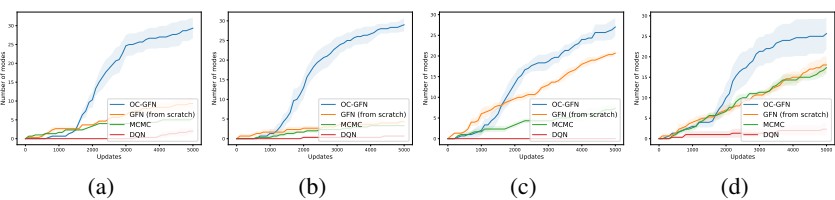

| | (a) | (b) | (c) | (d) |

Figure 20: Full Results in the RNA generation task.

## D LIMITATIONS AND FUTURE DIRECTIONS

We address the potential challenge of sparse rewards in training outcome-conditioned GFlowNet in the unsupervised pre-training stage with a contrastive learning procedure based on the concept of goal relabeling. This work lays the foundation for further exploration of pre-training and fine-tuning strategies in GFlowNets, enabling the transfer of pre-trained models to downstream tasks with different reward functions. However, it is also worth noting that Eq. (6) assumes that the state and action spaces remain consistent across many relevant problems. Extending our results to problems with different state and action spaces would be an exciting direction for future research. It is an interesting future direction for considering smoother rewards or other techniques for addressing this problem. In addition, it is also promising to consider continuous outcomes in future works (which has the potential to tackle even larger-scale problems), while our work mainly focuses on discrete outcomes.

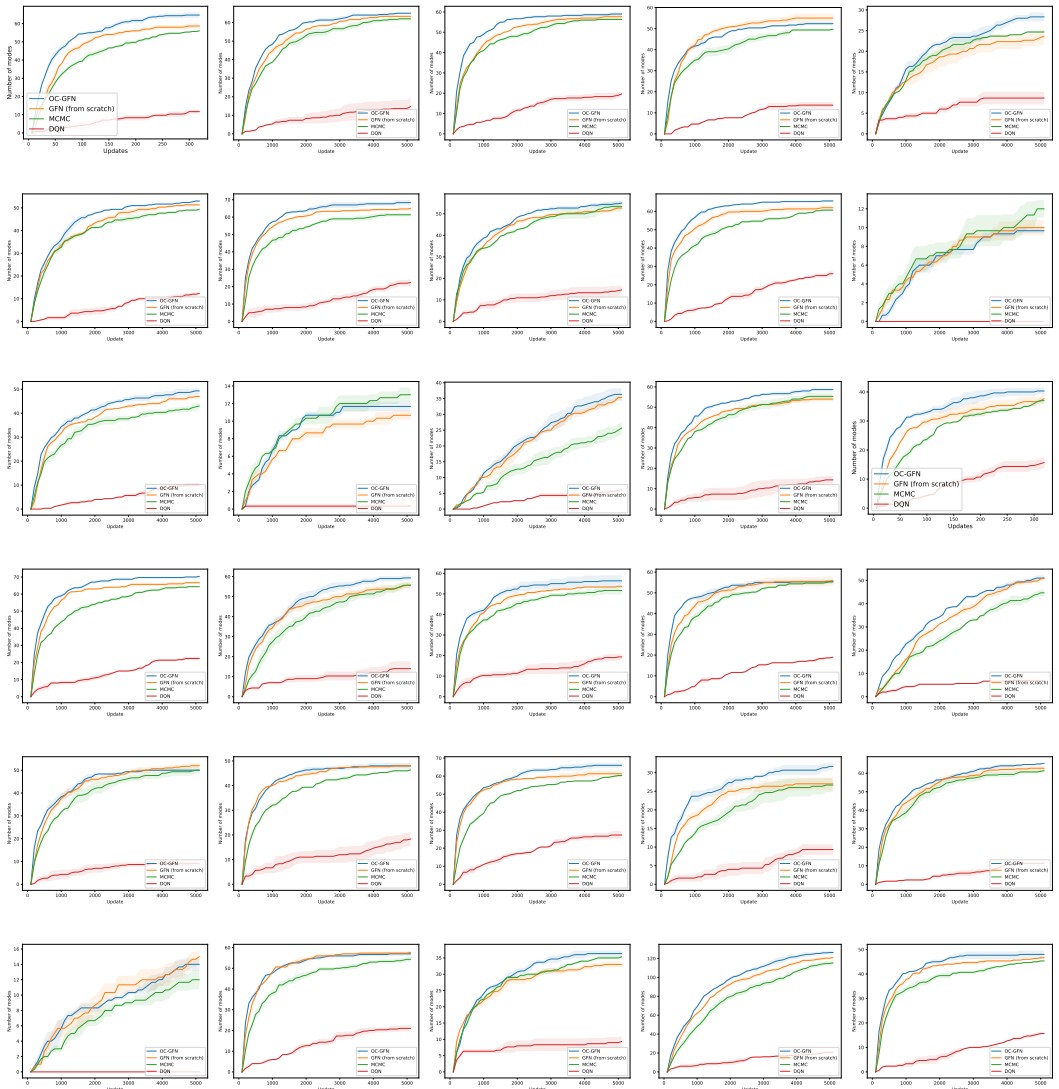

Figure 21: Full Results in the TF Bind generation task.

