# OpenReview forum: "Pre-Training and Fine-Tuning Generative Flow Networks"
_ICLR.cc/2024/Conference — ICLR 2024 spotlight_

### Official Review · Reviewer_gG5W · 2023-10-30

**Soundness:** 3 good
**Presentation:** 3 good
**Contribution:** 3 good
**Rating:** 8
**Confidence:** 4

**Summary:**

The paper describes a methodology for self-supervised reward-free pretraining of generative flow networks (GFlowNets). The authors propose a novel scheme for pretraining of GFlowNets and show its efficiency on a number of tasks. They also compare it with the baselines on a number of datasets. The authors include a number of improvements over the standard GFlowNets:

- Self-supervised pretraining

- Target task finetuning (or should it be called, e.g., task transfer as it does not finetune the coefficients of the original model to the new task?)

- Amortisation procedure for the target task finetuning to alleviate the problem of high cost of estimation of the task-specific reward function (Eq 6)

**Strengths:**

- (Originality, Significance) A multifold methodological contribution (see above), which helps define new ways to train and use the GFlowNet models, most importantly, including the insight about transferring the model to the downstream task without re-training (Section 4.2)

- (Quality) Well-written paper

- (Quality) Thorough analysis of the method on a number of tasks

- (Reproducibility/Clarity) The paper appears to provide good explanation of the experimental conditions and therefore, addresses reproducibility well (apart from Q2)

**Weaknesses:**

- (Soundness aspects) There are some questions to resolve about the motivation of the outcome teleportation module (see Q1), hence the current score.

- Clarifications on the experimental setting (see Q2)

**Questions:**

Q1:
While the experimental results show clear improvement when using the outcome teleportation module, the theoretical  motivation of Eq 4 still remains not entirely clear. The  original detailed balance equation represents the reversibility of the Markov chain; the proposed method, in contrast, does not satisfy such condition as it is seemingly assymetrical with backwards flow. One possible way would be to consider the right hand side a factorisation of the transition function $\tilde{P}_B(s | s’, y) = P_B (s | s’, y) R(x|y)$ but that won’t give $\int \tilde{P}_B(s | s’, y) ds = 1$. One can interpret it that it works as a regularisation of the loss function in Eq (5). Another related question relates to the transition between Eq 14 and Eq 15 in the Appendix related to the proof of this statement. It is not clear where did $R(x|y)$ disappear in between  Eq. (14) and (15). In the standard case of the detailed balance equation, that would have been a valid transition, but why is it valid for the non-1 R(x|y)?

Q2: Not sure I completely understand how the number of discovered modes (normalised and unnormalised) is calculated  (see Figures 10 and 11)?

Q3: “A remarkable aspect of GFlowNets is the ability to demonstrate the reasoning potential in generating a task-specific policy.“ Not sure the word reasoning would be the right way to describe it as I am not sure it is objective; despite not necessarily agreeing with the wording, I do think the meaning behind it, i.e. transferring to the task without re-training (Section 4.2), is a valuable aspect.

---

> ### Author Response · Authors · 2023-11-19
> **Response to Reviewer gG5W (Part 1 of 2)**
>
> We thank the reviewer for the valuable feedback on our work, and for noting that we propose "a novel scheme" that "defines new ways to train and use the GFlowNet models" with "insights about task tranfer" and "thorough analysis"， and show its "efficiency on a number of tasks". We greatly appreciate your comments!
>
> We have updated the paper to address the reviewer's comments, where the revisions are marked in red.
>
> *Q1: While the experimental results show clear improvement when using the outcome teleportation module, the theoretical motivation of Eq 4 still remains not entirely clear. The original detailed balance equation represents the reversibility of the Markov chain; the proposed method, in contrast, does not satisfy such condition as it is seemingly assymetrical with backwards flow. One possible way would be to consider the right hand side a factorisation of the transition function $\tilde{P}_B(s|s’,y) = P_B(s|s’,y)R(x|y)$ but that won’t give $\int \tilde{P}_B(s|s’,y)ds=1$. One can interpret it that it works as a regularisation of the loss function in Eq (5). Another related question relates to the transition between Eq 14 and Eq 15 in the Appendix related to the proof of this statement. It is not clear where did $R(s|y)$ disappear in between Eq. (14) and (15). In the standard case of the detailed balance equation, that would have been a valid transition, but why is it valid for the non-1 $R(x|y)$?*
>
> We have included additional derivation steps and clarified this in Appendix B.1 in the updated draft.
>
> (Regarding the motivation for Eq. (4): efficient credit assignment) The outcome teleportation technique we introduce in Section 4.1.1 is designed specifically for the outcome-conditioned GFlowNet (OC-GFN), which has a binary reward function -- the reward is $1$ when the outcome is reached while the reward is $0$ when the agent fails to reach the target outcome. In particular, Eq. (4) is valid only in the case of binary rewards and is designed for efficiently training OC-GFN -- incorporating the reward in the learning objective for each transition enables more efficient credit propagation.
>
> Let's consider the case when the OC-GFN is trained perfectly, where the success rate for reaching the given goal is $100\%$. If the outcome $y$ is a reachable goal from $s'$ (i.e., $R(x|y)=1$), then the flow consistency constraint in Eq. (4) will be $F(s, y) P_F(s'|s, y) = F(s'|y) P_B(s|s',y)$, which corresponds to usual detailed balance equation. Otherwise (when $R(x|y)=0$ for unreachable outcomes $y$), the flow consistency constraint will be $F(s,y) P_F(s'|s,y) = 0$, which quickly propagates the failure learning signal back to each transition directly.
>
> Another perspective is that it can be considered as a special kind of return/energy decomposition [1] for improving credit assignment, i.e., $R(x) = \prod_{t=0}^{n-1} r_t$, with $r_t$ denoting intermediate rewards. It can be considered as we decompose the terminal reward $1$ (or $0$) back to each step with a constant intermediate reward of $1$ (or $0$).
>
> (Regarding the proof) Accordingly in the proof, we leverage the fact that the reward for outcome-conditioned GFlowNet is binary ($0/1$) in Eq. (14) and Eq. (15) (which correspond to Eq. (17) and Eq. (18) in the updated draft). We first consider the case where the outcome is achieved, i.e., $x=y$ and $R(y|y)=1$ in Eq. (17), and then consider the other case where $R(x|y)= 0$ right after in Eq. (18).
>
> *Q2: Not sure I completely understand how the number of discovered modes (normalised and unnormalised) is calculated (see Figures 10 and 11)?*
>
> We use the same evaluation metrics as in the literature in GFlowNets [1-2]. The number of modes is calculated using a sphere-exclusion procedure. Specifically, the generated candidates are sorted by reward, and then scanning down this sorted list, a candidate is added to the list of modes if it is above a certain reward threshold and is further away than some distance threshold from all other modes. For the normalized number of modes, we normalize the number of modes discovered in each task by min-max normalization, and average this value across different tasks. This procedure is adapted from [1-2] and we have added this description in Appendix C.1.

---

> ### Author Response · Authors · 2023-11-19
> **Response to Reviewer gG5W (Part 2)**
>
> *Q3: “A remarkable aspect of GFlowNets is the ability to demonstrate the reasoning potential in generating a task-specific policy.” Not sure the word reasoning would be the right way to describe it as I am not sure it is objective; despite not necessarily agreeing with the wording, I do think the meaning behind it, i.e. transferring to the task without re-training (Section 4.2), is a valuable aspect.*
>
> We would like to thank the reviewer for recognizing the value of Section 4.2. Our goal was to emphasize the brain-inspired reasoning potential [3] that GFlowNets exhibit, while our work takes a step further by enabling efficient fine-tuning on downstream tasks. We acknowledge that the term "reasoning" may not be the most precise choice, and we have revised our wording to avoid confusion.
>
> ---
> We thank the reviewer for the time and effort in reviewing our work! We would greatly appreciate it if the reviewer could check our responses and the updates in the paper and let us know whether your concerns have been adequately addressed. We are happy to provide further clarification if you have any additional concerns.
>
> ---
> *References*
>
> [1] Pan, Ling, Nikolay Malkin, Dinghuai Zhang, and Yoshua Bengio. "Better training of gflownets with local credit and incomplete trajectories." arXiv preprint arXiv:2302.01687 (2023).
>
> [2] Bengio, Emmanuel, Moksh Jain, Maksym Korablyov, Doina Precup, and Yoshua Bengio. "Flow network based generative models for non-iterative diverse candidate generation." Advances in Neural Information Processing Systems 34 (2021): 27381-27394.
>
> [3] Bengio, Yoshua, Nikolay Malkin, and Moksh Jain. The GFlowNet Tutorial. https://milayb.notion.site/The-GFlowNet-Tutorial-95434ef0e2d94c24aab90e69b30be9b3, 2022.

---

> ### Comment · Reviewer_gG5W · 2023-11-20
> **A small clarification on the statement**
>
> Dear authors,
>
>
> As I am going through the responses, I would like to kindly ask to clarify on the following statement: "We would like to thank the reviewer for recognizing the value of Section 4.2. Our goal was to emphasize the brain-inspired reasoning potential [3] that GFlowNets exhibit, while our work takes a step further by enabling efficient fine-tuning on downstream tasks. We acknowledge that the term "reasoning" may not be the most precise choice, and we have revised our wording to avoid confusion."
>
> In the response, I see references [1], [2], [5] but not [3]. I believe it's a typo, but still think it needs clarity. What paper do you refer to?
>
> Many thanks,
> the Reviewer

---

> > ### Author Response · Authors · 2023-11-20
> > **Clarification**
> >
> > That is indeed a typo, apologies for the confusion. We were referring to "The GFlowNet Tutorial", in particular to the section "GFlowNets for Brain-Inspired Reasoning". We have fixed the reference number.

---

> > ### Author Response · Authors · 2023-11-21
> >
> > Dear Reviewer gG5W,
> >
> > We would like to thank you again for the detailed evaluation and helpful feedback! We would like to also thank you for going through our rebuttal!
> >
> > Since we are near the end of the discussion phase, we would like to take this opportunity to follow up on our previous responses. Please let us know if we have adequately addressed your concerns, or if there were any additional issues that we could address. We would be more than happy to discuss them during the ongoing discussion phase.
> >
> > Thank you for your time and effort!

---

> > > ### Comment · Reviewer_gG5W · 2023-11-21
> > >
> > > Dear Authors,
> > >
> > > I've gone through the responses to myself and other reviewers. Happy to change the score according to the revisions.
> > >
> > > Q1: Text before eq. (17): *Combining* Eq. 13. Other than that, I am happy with the clarification on the derivation, although I would clearly emphasise after Eq (4) that this is enabled by the fact the rewards are binary.
> > >
> > > Q2: Many thanks, it answers the original question
> > >
> > > Q3: The answer is clear, and changing the reasoning with adaptability looks good to me.
> > >
> > > I've also checked that there was another important comment from Reviewer ja74:
> > > "Q1: There should be a discussions of assumptions behind the OC-GFNs pretraining. Namely, that transfer is only possible when the reward function changes but not if the action-space or the state-space change. Moreover, the goal-conditioning requires a well specified set of outcomes Y — presumably not all states s are terminal states — which makes the proposed method not truly unsupervised. These limitations (together with the applicability mentioned at the end of A.2) could be stated explicitly in the main text, and left to future work."
> > > I agree that the assumptions behind OC-GFNs pretraining needed to be addressed, and I think the revised version covers this point as well.

---

> > > > ### Author Response · Authors · 2023-11-22
> > > >
> > > > Dear Reviewer gG5W,
> > > >
> > > > Thank you for your time and effort in reviewing our response and revisions, which have helped to improve our paper! We have also updated the draft to reflect your suggestions for Q1.
> > > >
> > > > We greatly appreciate your feedback and are glad that the changes have addressed your concerns. Thank you once again for your valuable suggestions and comments!

---

### Official Review · Reviewer_ja74 · 2023-11-01

**Soundness:** 3 good
**Presentation:** 3 good
**Contribution:** 3 good
**Rating:** 6
**Confidence:** 4

**Summary:**

- This paper tackles the problem of pretraining Generative Flow Networks (GFNs) and fine-tuning them to quickly approximate new sampling distributions.
- The authors take a reinforcement learning (RL) perspective, and observe that for GFNs the state-space, action-space, and transition probabilities remain unchanged for many problems of interests.
- This lets them apply goal-conditioned RL methods as a generic strategy to pretrain GFNs, which they dub outcome-conditioned GFNs (OC-GFNs).
- For fine-tuning, they show how to immediately adapt OC-GFNs when given the reward function of a downstream task (see Eq. 6). Since this involves the computation of an intractable sum, they amortize it with a learned predictor.
- The authors demonstrate the efficacy of their fine-tuned OC-GFNs on toy (GridWorld & BitSequence) and real-world biology problems (DNA binding, RNA generation, AMP generation), with some ablations on the toy problems.

**Strengths:**

- The exposition is generally clear, and I enjoyed reading the paper. The authors first present the goal-conditioning idea and how it applies to GFNs, then walk the reader through their derivation and assumptions for amortized adaptation. I especially appreciated Section 2 which gave a clear and concise background.
- The paper tackles an impactful problem for GFNs. While the pretraining solution is not particularly novel, it’s a neat application of goal-condition RL to an amortized sampling problem. The authors also figured out how to make it work on a wide range of problems, and provide several ablations in the main text and the appendix.
- The insight that a new sampling policy can be readily obtained from an outcome-conditioned flow is neat and, as far as I can tell, novel. This could spawn interest in outcome-conditioned flows and different ways to amortize Eq. 6.

**Weaknesses:**

- There should be a discussions of assumptions behind the OC-GFNs pretraining. Namely, that transfer is only possible when the reward function changes but not if the action-space or the state-space change. Moreover, the goal-conditioning requires a well specified set of outcomes Y — presumably not all states s are terminal states — which makes the proposed method not truly unsupervised. These limitations (together with the applicability mentioned at the end of A.2) could be stated explicitly in the main text, and left to future work.
- While there are enough benchmarks, I believe none include continuous action/state spaces. Moreover, the experiments only one GFN variant — the detailed-balance one, which is also used for OC-GFN. It would help validate the generality of OC if we had experiments showing it worked on these different settings. Moreover, I’d be curious to know how other pretrained amortized sampling baselines (eg, VAEs, normalizing flows) fare against OC-GFN — and what about pretraining a GFN on task A (without OC) and fine-tuning it on task B?
- (minor) The second and fourth paragraphs of Section 4.2 mention the “reasoning potential” of GFNs, and that intractable marginalization leads to “slow thinking”. Are these anthropomorphisms really needed for this paper?
- (minor) I wished the preliminaries (Section 2) included a training objective like Eq. 5 & 9, and that these more clearly specified which are the optimization variables.
- Some typos, there maybe more:
    - p. 3: multi-objective what?
    - p. 4: “given a reward R a posterior as a function”
    - p. 4: autotelicly → autotelically?
    - p. 5: “in log-scale obtained from Eq. (5)” should be Eq. 4?

**Questions:**

- Please comment on the weaknesses outlined above.
- Figures 10 and 11, right: Why is adaptation slower for OC-GFN than GFN in the first few thousand iterations? This is surprising since one would hope pretraining helps bootstrap downstream performance as in vision / language / RL. If it’s an exploration phase, did you validate it and is there a way to side-step it?

---

> ### Author Response · Authors · 2023-11-19
> **Response to Reviewer ja74 (Part 1 of 2)**
>
> We thank the reviewer for the useful feedback and positive assessment of our work, and for noting that we tackle "an impactful problem" with "a novel and neat insight", and "the exposition is generally clear". We are glad that the reviewer enjoyed reading the paper.
>  We greatly appreciate your comments!
>
> We seek to address each of your concerns as below, and we have updated the paper to address the reviewer's comments, where the revisions are marked in red.
>
> *Q1: There should be a discussions of assumptions behind the OC-GFNs pretraining. Namely, that transfer is only possible when the reward function changes but not if the action-space or the state-space change. Moreover, the goal-conditioning requires a well specified set of outcomes Y — presumably not all states s are terminal states — which makes the proposed method not truly unsupervised. These limitations (together with the applicability mentioned at the end of A.2) could be stated explicitly in the main text, and left to future work.*
>
> Thanks for the suggestion. For the definition of the terminal states, as in most GFlowNets benchmarks, the agent is allowed to take a "stop" action at every state, so every state can be a terminal state. We explicitly include this in Section 4 in the main text and Appendix D due to space limitation. We hope that our general framework can induce many fruitful follow-ups in the pre-training and fine-tuning paradigm of GFlowNets, and combine the recent success of this paradigm in vision, language, and RL domains.
>
> *Q2: While there are enough benchmarks, I believe none include continuous action/state spaces. Moreover, the experiments only one GFN variant — the detailed-balance one, which is also used for OC-GFN. It would help validate the generality of OC if we had experiments showing it worked on these different settings.*
>
> (Continuous state/action spaces) On the one hand, while continuous GFlowNets have recently been studied theoretically [1], in practice they remain limited in terms of applications. As noted in [2], continuous GFlowNets can be quite tricky to train and are challenging to scale to realistic tasks (which is also an open problem in GFlowNets). Therefore, we focus on the discrete case which is much more reliable (and most of the success benchmarks for GFlowNets focus on discrete space), and consider the practical and challenging tasks in the biological sequence generation domain. On the other hand, we do note that our method, in theory, is still applicable to the continuous case.
>
> (Applicability on other GFlowNets objectives) We include an additional study of OC-GFN when built upon the sub-trajectory balance [3] objective (OC-SubTB) in Section 5.1, where a detailed discussion can also be found in Appendix B.2. The experiments validate the versatility of our proposed methodology, which is general and can be applied to different GFlowNets learning objectives that learn state flow functions.
>
> *Q3: Moreover, I’d be curious to know how other pretrained amortized sampling baselines (eg, VAEs, normalizing flows) fare against OC-GFN.*
>
> Approaches such as VAEs and normalizing flows differ from GFlowNets in a key way -- they are learned using samples from some distribution and typically cannot be trained directly with a reward function. Although methods like MCMC can be used to sample proportionally to a given function, they typically do not take advantage of machine learning, and we are not aware of any such methods exploiting unsupervised pre-training for MCMC. While we are not aware of any such approaches that are applicable to the domains we study, we would be happy to include any such baselines for comparison.
>
> *Q4: What about pretraining a GFN on task A (without OC) and fine-tuning it on task B?*
>
> We include an additional comparison of a direct pre-training and fine-tuning scheme for GFlownets in Appendix C.3 (by pre-training a GFlowNet on task A, and then fine-tuning it on task B), which is also a motivation for our work. Results show that this simple scheme for pre-training GFlowNets does not lead to a universally helpful strategy for efficient transfer. This is because the GFlowNet learns to sample proportionally from the reward function of task A during the "pre-training" stage, and thus, if there is not enough shared structure between the rewards for task A and task B, the GFlowNet model pre-trained on task A may not be beneficial for the "fine-tuning" stage on task B, and could potentially even have a negative impact. On the contrary, our insight for unsupervised pre-training is that we cast this challenging problem as a self-supervised problem by learning to reach any goal for learning a *functional* understanding of the task.

---

> > ### Author Response · Authors · 2023-11-19
> > **Response to Reviewer ja74 (Part 2)**
> >
> > *Q5: (minor) The second and fourth paragraphs of Section 4.2 mention the "reasoning potential" of GFNs, and that intractable marginalization leads to “slow thinking”. Are these anthropomorphisms really needed for this paper?*
> >
> > We discuss this aspect to distinguish the pre-trained GFlowNet model from traditional RL policies, which typically learn a reward-maximizing policy and do not have this potential. Our goal was to emphasize the brain-inspired reasoning potential [4] that GFlowNets exhibit, while our work takes a step further by enabling efficient fine-tuning on downstream tasks. We acknowledge that the term "reasoning" may not be the most precise choice, and we have revised our wording to avoid confusion.
> >
> > *Q6: (minor) I wished the preliminaries (Section 2) included a training objective like Eq. 5 & 9, and that these more clearly specified which are the optimization variables.*
> >
> > Thanks for the suggestion, we did not include the corresponding loss function for the learning objective due to space limitation. We have now included the loss functions and optimization variables for Section 2 in Appendix A.1 for improved readability.
> >
> > *Q7: Some typos.*
> >
> > Thank you for your careful reading, and we have corrected the typos in the revision.
> >
> > *Q8: Figures 10 and 11, right: Why is adaptation slower for OC-GFN than GFN in the first few thousand iterations? This is surprising since one would hope pretraining helps bootstrap downstream performance as in vision / language / RL. If it’s an exploration phase, did you validate it and is there a way to side-step it?*
> >
> > Yes, our hypothesis is that it is an exploration phase in OC-GFN for warming up the amortized predictor (the numerator network $N$ and the outcome-sampling networks $Q$). We are currently actively exploring methods to accelerate the initial adaptation process, and it is also an interesting future direction to consider alternative and more sophisticated approaches for the amortized predictor.
> >
> > ---
> >
> > We thank the reviewer for the time and effort in reviewing our work! We would greatly appreciate it if the reviewer could check our responses and the updates in the paper and let us know whether your concerns have been adequately addressed. We are happy to provide further clarification if you have any additional concerns.
> >
> > ---
> >
> > *References*
> >
> > [1] Lahlou, Salem, Tristan Deleu, Pablo Lemos, Dinghuai Zhang, Alexandra Volokhova, Alex Hernández-Garcıa, Léna Néhale Ezzine, Yoshua Bengio, and Nikolay Malkin. "A theory of continuous generative flow networks." In International Conference on Machine Learning, pp. 18269-18300. PMLR, 2023.
> >
> > [2] Jain, Moksh, Tristan Deleu, Jason Hartford, Cheng-Hao Liu, Alex Hernandez-Garcia, and Yoshua Bengio. "GFlowNets for AI-driven scientific discovery." Digital Discovery 2, no. 3 (2023): 557-577.
> >
> > [3] Madan, Kanika, Jarrid Rector-Brooks, Maksym Korablyov, Emmanuel Bengio, Moksh Jain, Andrei Cristian Nica, Tom Bosc, Yoshua Bengio, and Nikolay Malkin. "Learning GFlowNets from partial episodes for improved convergence and stability." In International Conference on Machine Learning, pp. 23467-23483. PMLR, 2023.
> >
> > [4] Bengio, Yoshua, Nikolay Malkin, and Moksh Jain. The GFlowNet Tutorial. https://milayb.notion.site/The-GFlowNet-Tutorial-95434ef0e2d94c24aab90e69b30be9b3, 2022.

---

> ### Author Response · Authors · 2023-11-21
>
> Dear Reviewer ja74,
>
> We would like to thank you again for the detailed evaluation and helpful feedback! Since we are near the end of the discussion phase, we would like to take this opportunity to follow up on our previous responses. Please let us know if we have adequately addressed your concerns, or if there were any additional issues that we could address. We would be more than happy to discuss them during the ongoing discussion phase.
>
> Thank you for your time and effort!

---

> ### Author Response · Authors · 2023-11-23
>
> Dear Reviewer ja74,
>
> We would like to thank you again for the detailed evaluation and helpful feedback! As we approach the end of the discussion phase, we would like to take this opportunity to follow up on our previous responses and revisions.
>
> In summary, the major revisions include: 1) We included additional experiments to demonstrate the versatility of our approach by building OC-GFN upon the sub-trajectory balance (SubTB) objective, which validates the generality of our proposed framework. 2) We have included a comparison with a naive scheme for direct pre-training and fine-tuning, which motivated our work. This comparison highlights the inefficiency of the naive approach in achieving efficient transfer for GFlowNets. In contrast, our approach to unsupervised pre-training is based on the insight that we can frame this challenging problem as a self-supervised task by learning to reach any outcome, which enables us to develop a functional understanding of the task.
>
> Could you please let us know if we have adequately addressed your concerns? Thank you for your time and effort!

---

### Official Review · Reviewer_Deqj · 2023-11-01

**Soundness:** 3 good
**Presentation:** 3 good
**Contribution:** 3 good
**Rating:** 8
**Confidence:** 3

**Summary:**

The paper introduces a novel approach to pretrain generative flow networks (GFlowNet) in a self-supervised manner, focusing on aligning input with target outcomes. When adapting to a downstream task, there's no need to re-train the GFlowNet; instead, outcomes are integrated using Monte Carlo sampling. The authors cleverly introduce an amortized predictor to overcome sampling challenges.

**Strengths:**

The concept presented in this paper is both simple and elegant. The unsupervised fine-tuning approach offers a significant contribution, adeptly addressing the training challenges associated with GFlowNet. Overall, the paper is well-structured and easy to follow, making it a valuable addition to the literature.

**Weaknesses:**

See questions.

**Questions:**

In the 'Discussion about applicability' section, the trajectory balance's inability to learn the stateflow function, and its subsequent inapplicability for converting a pre-trained GFlowNet on a new reward, is mentioned. Have the authors evaluated the sub-trajectory balance objective (as per Pan et al.) which does incorporate the state-flow function?

---

> ### Author Response · Authors · 2023-11-19
> **Response to Reviewer Deqj**
>
> We thank the reviewer for the useful feedback and positive assessment of our work, and for noting that we introduce "a novel approach" with "an elegant concept" that "offers a significant contribution" in "a well-structured" way, which makes it "a valuable addition to the literature". We greatly appreciate your comments!
>
> We seek to address each of your concerns as below, and we have updated the paper to address the reviewer's comments, where the revisions are marked in red.
>
> *Q1: In the 'Discussion about applicability' section, the trajectory balance’s inability to learn the stateflow function, and its subsequent inapplicability for converting a pre-trained GFlowNet on a new reward, is mentioned. Have the authors evaluated the sub-trajectory balance objective (as per Pan et al.) which does incorporate the state-flow function?*
>
> Thank you for the suggestion. We include an additional study of OC-GFN when built upon the sub-trajectory balance objective [1] (OC-SubTB) in Section 5.1, where a detailed discussion can be found in Appendix B.2. The experiments validate the versatility of our proposed methodology, which is general and can be applied to different GFlowNets learning objectives that learn state flow functions.
>
> ---
>
> We thank the reviewer for the time and effort in reviewing our work! We would greatly appreciate it if the reviewer could check our responses and the updates in the paper and let us know whether your concerns have been adequately addressed. We are happy to provide further clarification if you have any additional concerns.
>
> ---
>
> *References*
>
> [1] Madan, Kanika, Jarrid Rector-Brooks, Maksym Korablyov, Emmanuel Bengio, Moksh Jain, Andrei Cristian Nica, Tom Bosc, Yoshua Bengio, and Nikolay Malkin. "Learning GFlowNets from partial episodes for improved convergence and stability." In International Conference on Machine Learning, pp. 23467-23483. PMLR, 2023.

---

### Official Review · Reviewer_5Pd3 · 2023-11-04

**Soundness:** 3 good
**Presentation:** 3 good
**Contribution:** 3 good
**Rating:** 6
**Confidence:** 4

**Summary:**

The paper  proposes the outcome-conditioned GFlowNet (OC-GFN) for reward-free pre-training and fine-tuning of GFlowNets in order for efficient adaptation to downstream tasks. OC-GFN is learnt to reach any specified outcome, and an amortized predictor is learnt to approximate an intractable marginal required for fine-tuning. The paper provides extensive experimental results to validate the effectiveness of their proposed approach.

**Strengths:**

1. The paper introduces a novel approach for reward-free pre-training and fine-tuning of GFlowNets, which can serve as a foundation for further research of GFlowNet pretraining.
2. The paper provides a thorough description of the proposed approach, including the formulation of the problem, the training procedures, and the evaluation metrics. The experiments are well-designed and conducted, and the results are presented clearly.

**Weaknesses:**

1. The paper lacks a comparison with existing approaches for pre-trained models or goal-conditioned RL methods.

**Questions:**

1. How does the proposed approach perform compared to existing methods for pre-trained models or RL methods besides DQN? and what about the computation cost of these methods?
2. Is the trained GAFlowNet necessary? What about its performance and how does it influence the results?

---

> ### Author Response · Authors · 2023-11-19
> **Response to Reviewer 5Pd3 (Part 1 of 2)**
>
> We thank the reviewer for the useful feedback and positive assessment of our work, and for noting that we introduce "a novel approach" and provide "a thorough description" of the proposed approach with "well-designed and conducted experiments", where "extensive results are presented clearly", and our work "can serve as a foundation" for future research on GFlowNet pretraining. We greatly appreciate your comments!
>
> We seek to address each of your concerns as below, and we have updated the paper to address the reviewer's comments, where the revisions are marked in red.
>
> *Q1: How does the proposed approach perform compared to existing methods for pre-trained models or RL methods besides DQN? And what about the computation cost of these methods?*
>
> We would like to highlight that we propose the *first* pre-training paradigm for GFlowNets, and there are no existing methods for pre-training GFlowNets models to the best of our knowledge. However, as a point of comparison, we also include a discussion of a simple scheme (pre-train a GFlowNet on task A and then fine-tune it on task B) in Appendix C.3, which also serves as a motivation for our work. Results show that this simple scheme for pre-training GFlowNets does not lead to a universally applicable strategy for efficient transfer. This is because the GFlowNet learns to sample proportionally from the reward function of task A during the "pre-training" stage, and thus, if there is not enough shared structure between the rewards for task A and task B, the GFlowNet model pre-trained on task A may not be beneficial for the "fine-tuning" stage on task B, and could potentially even have a negative impact. On the contrary, our insight for unsupervised pre-training is that we cast this challenging problem as a self-supervised problem by learning to reach any outcome for learning a "functional" understanding of the task.
>
> Regarding other RL baselines, we include additional comparisons with two other strong RL baselines (PPO and SAC) besides DQN in Appendix C.4 on two downstream tasks in TF Bind generation due to time limitation for the rebuttal period. As shown, the reward-maximization nature of RL approaches limits the resulting solution diversity [1], where OC-GFN continues to outperform these baselines consistently.
>
> We also include computation costs for each method in Appendix C.4 (Figure 18), where we observe that OC-GFN only induces marginal additional computation cost.
>
> *Q2: Is the trained GAFlowNet necessary? What about its performance and how does it influence the results?*
>
> Thank you for the question. The GAFlowNet model plays a critical role in the unsupervised pre-training stage. Please find our detailed discussion and experiments below.
>
> (Intuition) GAFlowNet [2] enables the learning of GFlowNet driven by intrinsic rewards (e.g., curiosity about states [3] -- which gives a higher reward for less familiar/unseen states and vice versa). This makes it very suitable in the unsupervised pre-training stage [4] where the agent generates the outcomes autotelically, which requires training it with full support over the outcomes. GAFlowNet improves the efficiency of generating diverse outcomes, which is particularly important in higher-dimensional spaces.
>
> (Experiments) We include an ablation study for the effect of the GAFlowNet model by replacing it with a random agent and a GFlowNet which is trained with task-irrelevant rewards ($R(x)=1$) in the unsupervised pre-training stage. (Please also note that the effect of completely removing this part has been studied in Section 5.1 (Figure 4), which is a key to the success of the unsupervised pre-training stage.)
>
> (Empirical results) We compare the success rate of each method for reaching given outcomes following Section 5.1. As shown in Figure 14 in Appendix C.2, OC-GFN (GAFN) learns much more efficiently than the other two variants, where the GAFN model is replaced with a GFN model trained with task-irrelevant rewards (OC-GFN (GFN)) or a random agent (OC-GFN (random)), with a more significant margin in larger maps. In addition,  we also study the generated outcome distribution by ablating this model design to gain a better understanding during the learning process. Figure 15 in Appendix C.2 demonstrates the outcome distribution generated by each method at the same iteration in a large map. As shown, GAFN generates outcomes that allow the agent to explore unfamiliar regions, resulting in a more diverse outcome distribution for training the OC-GFN and improving its learning efficiency, which also demonstrates the importance of the GAFN model.

---

> > ### Author Response · Authors · 2023-11-19
> > **Response to Reviewer 5Pd3 (Part 2)**
> >
> > We thank the reviewer for the time and effort in reviewing our work! We would greatly appreciate it if the reviewer could check our responses and the updates in the paper and let us know whether your concerns have been adequately addressed. We are happy to provide further clarification if you have any additional concerns.
> >
> > ---
> >
> > *References*
> >
> > [1] Bengio, Emmanuel, Moksh Jain, Maksym Korablyov, Doina Precup, and Yoshua Bengio. "Flow network based generative models for non-iterative diverse candidate generation." Advances in Neural Information Processing Systems 34 (2021): 27381-27394.
> >
> > [2] Pan, Ling, Dinghuai Zhang, Aaron Courville, Longbo Huang, and Yoshua Bengio. "Generative Augmented Flow Networks." In The Eleventh International Conference on Learning Representations. 2022.
> >
> > [3] Burda, Yuri, Harrison Edwards, Amos Storkey, and Oleg Klimov. "Exploration by random network distillation." In International Conference on Learning Representations. 2018.
> >
> > [4] Laskin, Michael, Denis Yarats, Hao Liu, Kimin Lee, Albert Zhan, Kevin Lu, Catherine Cang, Lerrel Pinto, and Pieter Abbeel. "URLB: Unsupervised reinforcement learning benchmark." arXiv preprint arXiv:2110.15191 (2021).

---

> ### Author Response · Authors · 2023-11-21
>
> Dear Reviewer 5Pd3,
>
> We would like to thank you again for the detailed evaluation and helpful feedback! Since we are near the end of the discussion phase, we would like to take this opportunity to follow up on our previous responses. Please let us know if we have adequately addressed your concerns, or if there were any additional issues that we could address. We would be more than happy to discuss them during the ongoing discussion phase.
>
> Thank you for your time and effort!

---

> ### Author Response · Authors · 2023-11-23
>
> Dear Reviewer 5Pd3,
>
> We would like to thank you again for the detailed evaluation and helpful feedback! As we approach the end of the discussion phase, we would like to take this opportunity to follow up on our previous responses and revisions.
>
> In summary, the major revisions include: 1) We have included additional RL baselines (PPO and SAC) to provide a more comprehensive comparison, where the results show that OC-GFN continues to outperform these baselines consistently. 2) Additionally, we have expanded our discussion and conducted further experiments to demonstrate the importance of GAFlowNets in the unsupervised pre-training stage.
>
> Could you please let us know if we have adequately addressed your concerns? Thank you for your time and effort!

---

### Meta-Review · Area_Chair_Ahqn · 2023-12-19

**Metareview:**

The paper presents an approach to pre-training and fine-tuning GFlowNets via outcome conditioning. All reviewers appreciated innovation and detailed experimental validation, although also wished for more comprehensive comparative analysis. For these reasons, I recommend acceptance.

**Justification For Why Not Higher Score:**

Relative lack of comprehensive comparative analysis.

**Justification For Why Not Lower Score:**

Innovation and experimental validation.

---

### Decision · Program_Chairs · 2024-01-16

Accept (spotlight)